# Electrically driven lasing from a dual-cavity perovskite device

Chen Zou[1,2✉], Zhixiang Ren[1], Kangshuo Hui[1], Zixiang Wang[1], Yangning Fan[1], Yichen Yang[1], Bo Yuan[1], Baodan Zhao[1,2✉] & Dawei Di[1,2,3✉]

Solution-processed semiconductor lasers promise lightweight, wearable and scalable optoelectronic applications. Among the gain media for solution-processed lasers, metal halide perovskites stand out as an exceptional class because of their ability to achieve wavelength-adjustable, low-threshold lasing under optical pumping[1–8]. Despite the progress in this field, electrically driven lasing from perovskite semiconductors remains a critical challenge. Here we demonstrate an electrically driven perovskite laser, constructed by vertically integrating a low-threshold single-crystal perovskite microcavity sub-unit with a high-power microcavity perovskite LED (PeLED) sub-unit. Under pulsed electrical excitation, the dual-cavity perovskite device shows a minimum lasing threshold of 92 A cm$^{-2}$ (average threshold: 129 A cm$^{-2}$, at about 22 °C, in air), which is an order of magnitude lower than that of state-of-the-art electrically driven organic lasers[9,10]. Key to this demonstration is the integrated dual-cavity device architecture, which allows the microcavity PeLED sub-unit to deliver directional emission into the single-crystal perovskite microcavity sub-unit (at a coupling efficiency of about 82.7%) to establish the lasing action. An operational half-life ($T_{50}$) of 1.8 h (6.4 × 10$^4$ voltage pulses at 10 Hz) is achieved, outperforming the stability of electrically pumped organic lasers[9,10]. The dual-cavity perovskite laser can be rapidly modulated at a bandwidth of 36.2 MHz, indicating its potential for data transmission and computational applications.

Metal halide perovskites are an emerging class of semiconductors[11,12] combining remarkable optoelectronic properties with cost-effective solution processability. Apart from the rapid advances in solar cells[13–15] and LEDs[16–22], the unique attributes of high gain coefficients, long carrier lifetimes and tunable emission wavelengths have made perovskites excellent optical gain media for lasing applications[1–8]. Room-temperature, continuous wave lasing from perovskite semiconductors was demonstrated under optical pumping[4,23–26], highlighting the promising progress in the field.

The success of conventional semiconductor lasers builds on the ability of electrically driving the lasing action[27–29], allowing them to be easily integrated with a range of optoelectronic device platforms. However, for halide perovskites, the realization of electrically driven lasing remains a great challenge because of the inability to achieve intense electrical injection into high-quality perovskite resonant cavities. High currents can lead to severe material degradation and efficiency roll-off, whereas standard optical cavity designs are poorly compatible with the perovskite device architectures[30–33].

In this work, we demonstrate electrically driven lasing from a dual-cavity perovskite device, which integrates a low-threshold perovskite single-crystal microcavity sub-unit with a high-power microcavity PeLED sub-unit to form a vertically stacked multi-layer structure. Under pulsed electrical excitation, the dual-cavity perovskite device shows a low lasing threshold of about 92 A cm$^{-2}$ (at around 22 °C, in air), which is about 30 times lower than that recently reported for electrically driven integrated organic lasers (with a threshold of about 2.8 kA cm$^{-2}$) (ref. 10). The dual-cavity device architecture is essential for this demonstration. It allows the microcavity PeLED sub-unit to deliver concentrated optical power into the single-crystal perovskite microcavity sub-unit (at an inter-cavity optical coupling efficiency of 82.7%) to support the lasing action. The laser device shows an operational half-life ($T_{50}$) of 1.8 h (6.4 × 10$^4$ voltage pulses at 10 Hz), exhibiting improved durability compared with electrically driven organic lasers[9,10]. The dual-cavity perovskite laser can be rapidly modulated at a bandwidth of 36.2 MHz, indicating its possible applications in data transmission and computation.

## Structure of the integrated dual-cavity perovskite laser

The structure of the integrated dual-cavity device is shown in Fig. 1a. The two sub-units, each containing a microcavity, were heterogeneously integrated to form a multi-layer device stack. During device operation, the directional electroluminescence (EL) from the PeLED in the first microcavity (microcavity I) is absorbed by a perovskite single crystal in the second microcavity (microcavity II), which supports light

[1]State Key Laboratory of Extreme Photonics and Instrumentation, College of Optical Science and Engineering, International Research Center for Advanced Photonics, Zhejiang University, Hangzhou, China. [2]ZJU-Hangzhou Global Scientific and Technological Innovation Center, Zhejiang University, Hangzhou, China. [3]ZJU-UIUC Institute, Zhejiang University, Haining, China. ✉e-mail: zouchen@zju.edu.cn; baodanzhao@zju.edu.cn; daweidi@zju.edu.cn

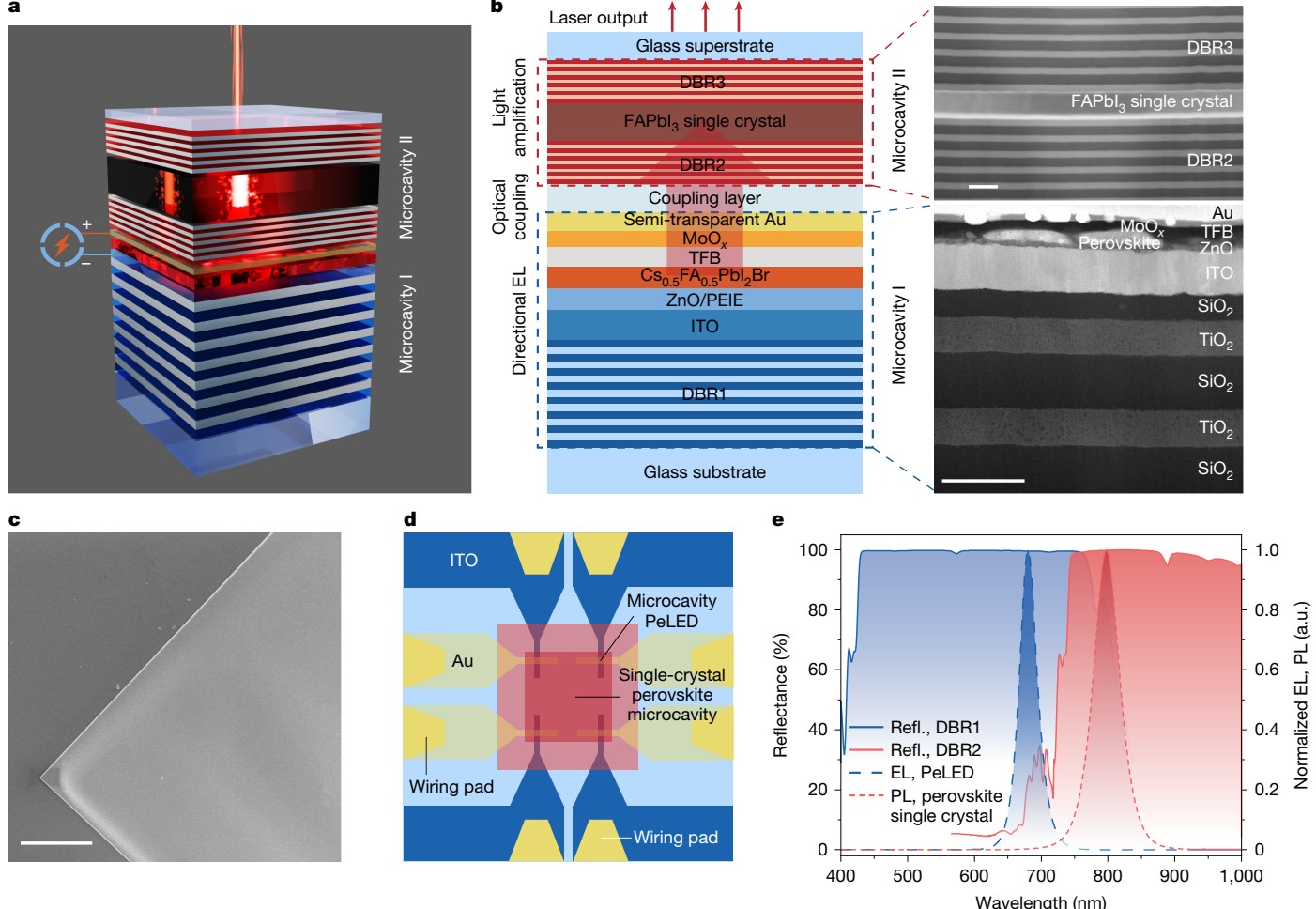

**Fig. 1 | Structure and basic optical properties of the integrated dual-cavity perovskite laser. a**, Schematic of the device. **b**, A cross-sectional schematic of the device structure (left), and high-angle annular dark-field scanning transmission electron microscopy (HAADF-STEM) images of a device (layer thicknesses unoptimized) (right). See Methods for detailed fabrication processes. **c**, A top-view SEM image of an FAPbI₃ single crystal used in microcavity II.

**d**, A schematic top view of the device. The Au and ITO electrodes are the anode and cathode, respectively. **e**, Reflection spectra of DBR1 in microcavity I and DBR2 in microcavity II, together with the EL spectrum of the $Cs_{0.5}FA_{0.5}PbI_2Br$ microcavity PeLED and the PL spectrum of the FAPbI₃ single crystal. Refl., refelction; a.u., arbitrary units. Scale bars, 400 nm (**b**, top right); 200 nm (**b**, bottom right); 10 µm (**c**).

amplification (Fig. 1a,b). The microcavity PeLED sub-unit has a structure of glass/distributed Bragg reflector (DBR)/indium tin oxide (ITO)/ZnO/polyethyleneimine ethoxylated (PEIE)/$Cs_{0.5}FA_{0.5}PbI_2Br$ perovskite/poly(9,9-dioctyl-fluorene-co-n-(4-butylphenyl) diphenylamine) (TFB)/$MoO_x$/Au (Fig. 1b). The DBR (DBR1) consists of alternating layers of $SiO_2$/$TiO_2$, featuring a high reflectance (>99%) at about 680 nm. The Au electrode is about 20-nm thick, showing semi-transparency with an optical reflectance of 45% at 680 nm (Supplementary Fig. 1). The cavity mode at around 680 nm can be supported by microcavity I (Supplementary Note 1). The perovskite single-crystal microcavity (microcavity II) was constructed by high-quality FAPbI₃ perovskite single crystals embedded in two high-reflectance (>98% at around 800 nm) DBRs (DBR2 and DBR3) (Fig. 1b). The typical size of the crystals is about 0.08 mm² (0.29 mm × 0.27 mm) (Supplementary Fig. 2). The perovskite single crystals were grown in situ between the DBRs through space-confined inverse-temperature crystallization[34–37] (Extended Data Fig. 1a) (Methods). The thickness of the single crystals can be controlled by the height of the gold spacer between the two DBRs[34,35], and was optimized to be approximately 180 nm. Scanning electron microscopy (SEM) images of the single crystals show a uniform surface morphology (Fig. 1c and Extended Data Fig. 1b). The root-mean-square roughness of the samples is about 0.7 nm (Extended Data Fig. 1c). Two-dimensional (2D) X-ray diffraction (XRD) results of the FAPbI₃ single crystals exhibit high

crystallinity (Extended Data Fig. 2a). The XRD patterns of the FAPbI₃ crystals show two sharp diffraction peaks corresponding to the (200) and (400) planes (Extended Data Fig. 2b), indicating that the perovskite is of cubic phase[36]. The single-crystal samples show sharper and fewer XRD peaks compared with the polycrystalline films with a similar composition, consistent with the higher crystallinity of the samples[37].

To improve the optical power density of the microcavity PeLED sub-unit, small active areas of down to 0.02 mm² (determined by the overlapping area of the ITO and Au electrodes) are used (Fig. 1d). Tapered electrodes are employed to allow good electrical conduction. The EL of the microcavity $Cs_{0.5}FA_{0.5}PbI_2Br$ PeLED sub-unit and the photoluminescence (PL) of the FAPbI₃ single crystals (without optical cavities) show spectral peaks at around 680 nm and 790 nm, respectively (Fig. 1e). The DBR paired with the semi-transparent gold electrode to form microcavity I (DBR1) exhibits high reflectance (>99%) from 450 nm to 750 nm. The bottom DBR in microcavity II (DBR2) shows high reflectance (>98%) at 750–1,000 nm and high transmittance (>90%) for wavelengths shorter than 680 nm. The top DBR in microcavity II (DBR3) features high reflectance (>98%) at 620–870 nm in optimized devices (Supplementary Fig. 3). It prevents the residual emission of microcavity I (the part that is not fully absorbed by the perovskite single crystals) from escaping from the top surface of the dual-cavity device. This dual-cavity configuration allows the emission from microcavity I

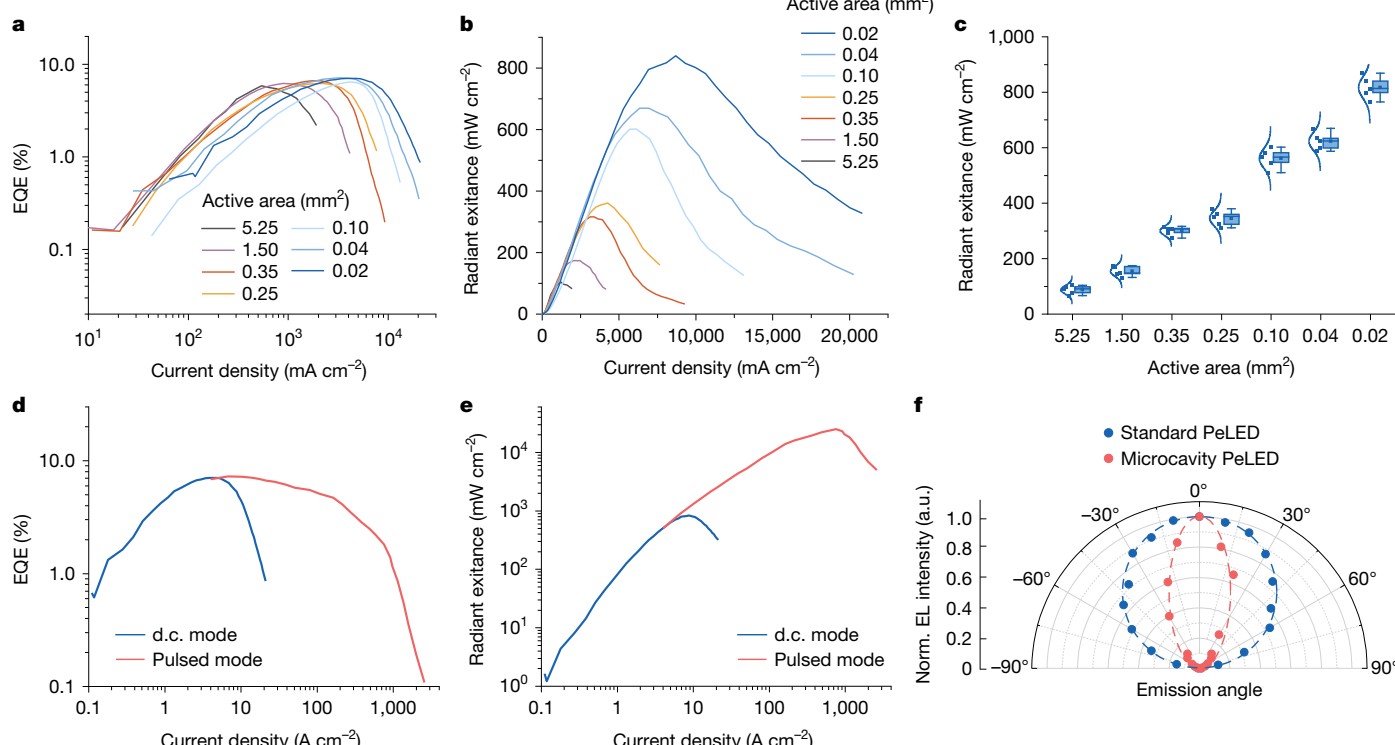

**Fig. 2 | Characteristics of the microcavity PeLED sub-unit. a**, EQE as a function of current density under d.c. operation. **b**, Radiant exitance as a function of current density under d.c. operation. **c**, Radiant exitance as a function of device area. The blue curves are the Gaussian fits to the data (five samples for each type). The horizontal lines in the box plots represent the 25th percentile, the median and the 75th percentile (from bottom to top). The bottom and top whiskers mark the 5th and 95th percentiles, respectively. The active area of each device is defined by the overlapping region of the ITO and Au electrodes (the width of ITO times the width of Au). The device areas tested were 5.25 mm² (3.5 mm × 1.5 mm), 1.5 mm² (1.5 mm × 1.0 mm), 0.35 mm² (1 mm × 0.35 mm), 0.25 mm² (0.5 mm × 0.5 mm), 0.1 mm² (0.5 mm × 0.2 mm), 0.04 mm² (0.2 mm × 0.2 mm) and 0.02 mm² (0.14 mm × 0.14 mm). **d**, EQE–current density data of a microcavity PeLED (0.02 mm²) under d.c. and pulsed operation. **e**, Radiant exitance–current density data of a microcavity PeLED (0.02 mm²) under d.c. and pulsed operation. **f**, Angular emission intensity profiles of standard and microcavity PeLEDs.

to be coupled with microcavity II, providing the optical power and feedback mechanism, which enable lasing from the FAPbI₃ single crystal.

## High-power microcavity PeLED sub-unit

Reducing the active area of the microcavity PeLED sub-unit (microcavity I) from 5.25 mm² to 0.02 mm² is beneficial for suppressing the EL efficiency roll-off at high current densities (Fig. 2a), leading to a high radiant exitance of up to about 868 mW cm⁻² under d.c. operation (Fig. 2b,c), corresponding to a high equivalent radiance of 7,032 W sr⁻¹ m⁻². The reasonably uniform EL intensity profile of the active area indicates the limited influence of current funnelling effects arising from narrow current pathways (Supplementary Fig. 4). Under pulsed operation (with a pulse duration of 1 µs and a repetition rate of 100 Hz; Extended Data Fig. 3), the microcavity PeLED sub-unit could stand current densities of up to 2.5 kA cm⁻² before device breakdown (Fig. 2d), resulting in a maximum radiant exitance of around 2.51 × 10⁴ mW cm⁻² (about 25.1 W cm⁻²) from the PeLED sub-unit (Fig. 2e). This corresponds to a maximum equivalent radiance of 2.03 × 10⁵ W sr⁻¹ m⁻², placing it among PeLEDs with the highest output power density[6,38].

The microcavity top-emission PeLEDs show similar EQE and radiant exitance compared with standard bottom-emission PeLEDs prepared using identical emissive and charge-transport materials (Extended Data Fig. 4). However, in contrast to standard PeLEDs with an output optical power distribution following the Lambertian profile, microcavity PeLEDs show a clear angular dependence of the cavity mode and are capable of delivering more convergent emission from microcavity I to microcavity II (Fig. 2f and Extended Data Fig. 5). The use of the

microcavity PeLED sub-unit for the effective coupling of optical power into the second cavity is a key design consideration that has enabled lasing, as we discuss below.

## Electrically driven dual-cavity perovskite laser

The dual-cavity perovskite device was driven by electrical pulses under room temperature in air (about 22 °C, about 50% relative humidity), with the emission spectra at different current densities recorded (Fig. 3a). At lower current densities of 56–85 A cm⁻², the device produced relatively weak emission with broad spectra corresponding to the cavity mode of the single-crystal perovskite microcavity. As the current density increased to about 119 A cm⁻², a narrow emission peak at around 803 nm emerged. The output light intensity of the dual-cavity device as a function of the pulsed current density is shown in Fig. 3b. The emission intensity increased sharply as the current density rose over a threshold of approximately 92 A cm⁻². At the same time, linewidth narrowing from about 1.03 nm to 0.44 nm across the threshold was observed (Fig. 3b).

The polarization properties of the emission from our dual-cavity perovskite laser device were measured (Fig. 3c). Below the threshold, the output intensity shows no correlation with polarization angle, suggesting the output light is unpolarized. Above the threshold, the emission from the device exhibits a clear linear polarization, consistent with the behaviour of lasing. The oriented linear polarization above the threshold possibly originates from structural asymmetry and material anisotropy[39,40]. Under optical pumping with a focused pump spot (about 18 µm diameter), different linear polarization configurations

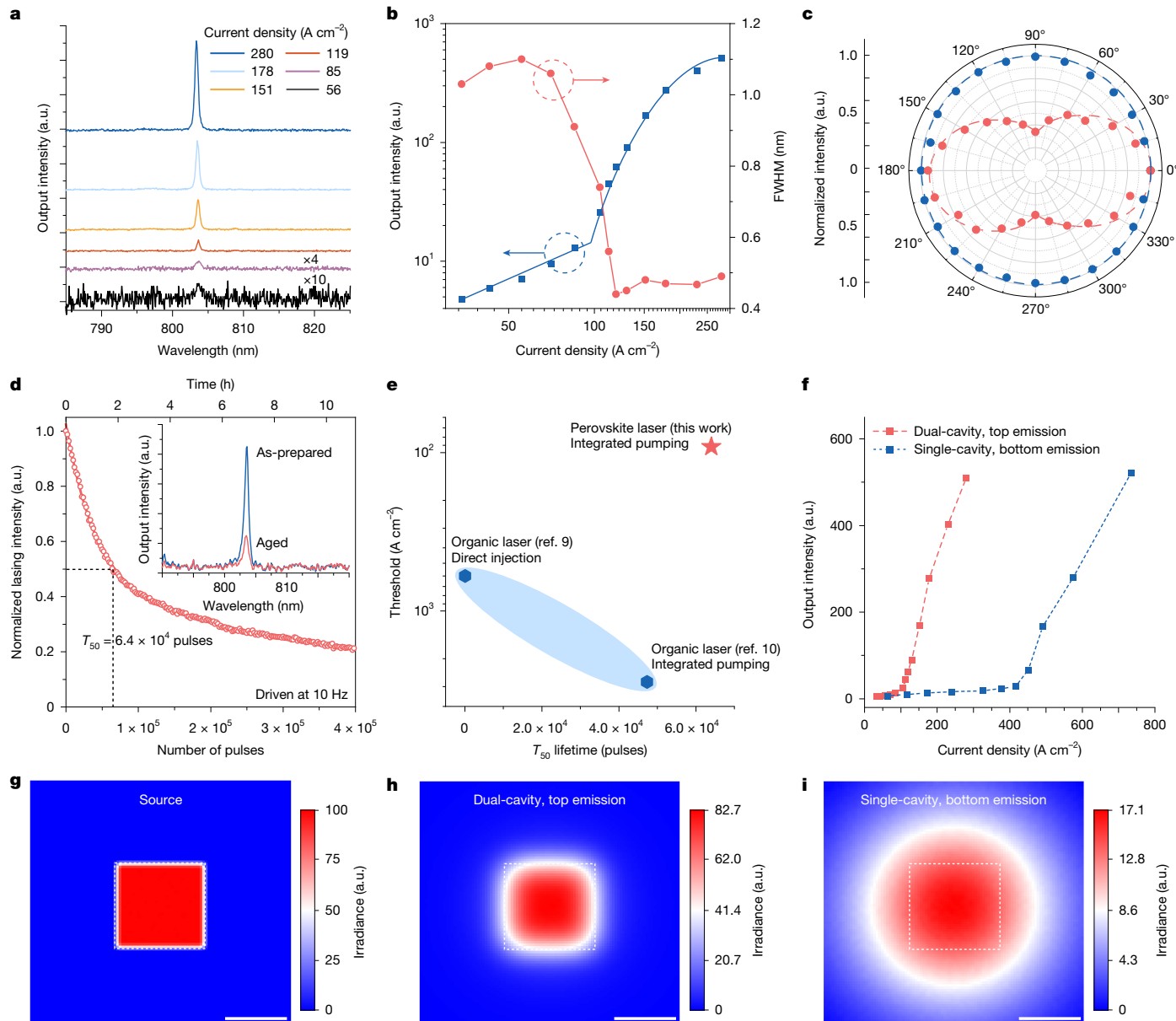

**Fig. 3 | Characterization of electrically driven dual-cavity perovskite lasers.** Measured in air (temperature about 22 °C and relative humidity about 50%). **a**, Emission spectra of the perovskite laser driven at different current densities. The intensities of the emission spectra at 56 A cm$^{-2}$ and 85 A cm$^{-2}$ (below threshold) are magnified by 10 and 4 times, respectively. **b**, Output intensity and the spectral FWHM compared with current density. **c**, Polarization plot of a dual-cavity perovskite laser driven at a current density of 280 A cm$^{-2}$ (above threshold, red) and 56 A cm$^{-2}$ (below threshold, blue). **d**, Operational stability test of an electrically driven dual-cavity perovskite laser (peak current density 163 A cm$^{-2}$, pulse width 1 μs and repetition rate 10 Hz). The inset shows the emission spectra of as-prepared (fresh) and aged devices. **e**, Thresholds and operational lifetimes of our electrically driven perovskite laser compared with the state-of-the-art electrically driven organic lasers. **f**, Output light intensity compared with current density for two types of perovskite lasers based on dual-cavity and single-cavity architectures, respectively. **g**, Simulated optical power distribution in the PeLED active area (source) in microcavity I/PeLED, with the peak power set to 100. **h**,**i**, Simulated optical power (irradiance) distributions in microcavity II, when a dual-cavity device structure featuring a microcavity top-emission PeLED sub-unit (**h**) and a single-cavity device structure featuring a standard bottom-emission PeLED sub-unit (**i**) are used. The boxed region represents the active area of the source in microcavity I. a.u., arbitrary units. Scale bar, 100 μm (**g–i**).

(for example, orientation and degree of polarization) of microcavity II were observed at different lasing spots (Supplementary Fig. 5). The orientations of linear polarization of the electrically driven lasing from our dual-cavity perovskite devices exhibit device-to-device variation (Supplementary Fig. 6), consistent with the observations in other classes of vertical-cavity lasers[41,42]. The moderate polarization observed probably arises from an average effect. The far-field images (Extended Data Fig. 6) of the emission beam below and above the threshold were recorded using a CCD camera. The emission beam profile shows a transition from weak diffuse luminescence to intense directional emission (with a divergence angle of about 3.7°), consistent with the transition from spontaneous emission to lasing. Together, the clear threshold (Fig. 3a), the linewidth narrowing (Fig. 3b), the polarization properties (Fig. 3c) and the beam profile (Extended Data Fig. 6) demonstrate lasing from the dual-cavity perovskite device, according to established measurement protocols[43].

We measured the operational stability of our dual-cavity lasers under electrical excitation (current density 163 A cm$^{-2}$, pulse width 1 μs and repetition rate 10 Hz). The $T_{50}$ lifetime (the time required for the emission intensity to drop to half of the initial intensity) of the

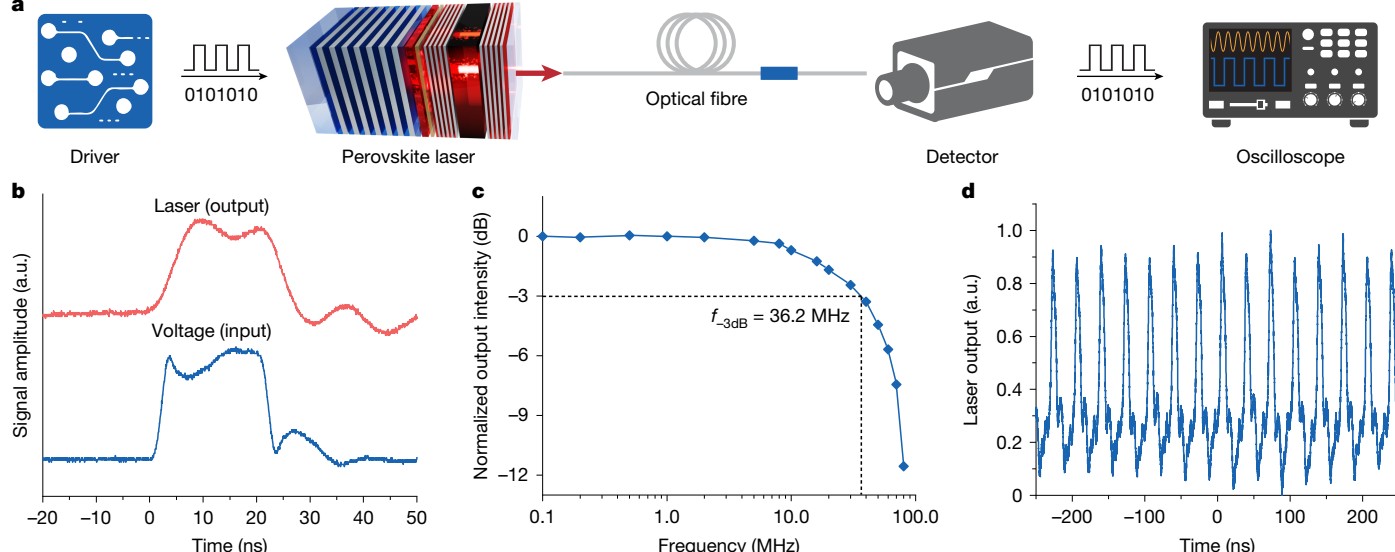

**Fig. 4 | Frequency response of the electrically driven dual-cavity perovskite laser. a**, Schematic of the experimental setup. **b**, Transient profiles of the driving voltage pulses and the laser pulses (peak voltage 50 V, pulse duration 20 ns and repetition rate 100 Hz). **c**, Frequency response of the electrically driven dual-cavity perovskite laser. **d**, Output lasing pulses from a dual-cavity perovskite laser operating at 30 MHz (driving voltage 50 V and pulse width 5 ns). The measurements were performed at room temperature in air (temperature about 22 °C and relative humidity about 50%). a.u., arbitrary units.

perovskite laser is 1.8 h, corresponding to $6.4 \times 10^4$ pulses (Fig. 3d). The lasing process lasted for more than $4 \times 10^5$ consecutive pulses, without any observable changes in emission peak wavelength and linewidth (Fig. 3d, inset). The results indicate that our dual-cavity perovskite lasers can offer lower thresholds and improved durability over the state-of-the-art electrically driven organic lasers[9,10] (Fig. 3e).

The output power density of the microcavity PeLED sub-unit at 92 A cm$^{-2}$ is 8.9 W cm$^{-2}$, only slightly higher than the lasing threshold (6.6 W cm$^{-2}$) of microcavity II under quasi-continuous wave optical pumping (Extended Data Fig. 7). This is consistent with the efficient coupling between the two microcavities we discussed earlier. The low lasing thresholds of microcavity II under quasi-continuous wave optical pumping are reproducible (Extended Data Fig. 7d) and are consistent with the low lasing thresholds (about 0.47 μJ cm$^{-2}$) under femtosecond optical pumping and the long carrier lifetime (around 134.7 ns) at the excitation density ($3.9 \times 10^{16}$ cm$^{-3}$) corresponding to the lasing threshold (Extended Data Fig. 8 and Supplementary Note 2). The dual-cavity perovskite device shows a much lower (around 4.7 times) minimum lasing threshold (92 A cm$^{-2}$) than those based on an integrated single-cavity device featuring a standard PeLED (433 A cm$^{-2}$) (Fig. 3f and Extended Data Fig. 9). The average lasing threshold of the dual-cavity perovskite devices is 129 A cm$^{-2}$, with a standard deviation of 27 A cm$^{-2}$. We consider that the slightly divergent emission from microcavity I may expand to form a broader beam when travelling a longer distance, potentially reducing the inter-cavity coupling efficiency and the power density for excitation[44,45]. This indicates the importance of the directional (less divergent) emission as well as the vertically compact (short-distance) integration, both enabled by the dual-cavity architecture. Simulation results indicate that the inter-cavity coupling efficiency improves from 17.1% (with a standard bottom-emission PeLED) to 30.2% (with a microcavity top-emission PeLED) at a relatively long coupling distance of 350 μm (Supplementary Fig. 7). This coupling efficiency is further enhanced to 82.7% by reducing the coupling distance from 350 μm to 50 μm. As a result, our microcavity PeLED sub-unit delivers substantially higher (4.84 times) optical power density to the second microcavity compared with a standard PeLED of the same power (Fig. 3g–i), consistent with the strong reduction of the lasing threshold.

## Modulation properties of the perovskite laser

The modulation properties of our perovskite laser were tested using a signal transmission setup (Fig. 4a). The perovskite laser was electrically modulated by voltage pulses, generating binary (0 and 1) digital signals. The laser beam emitted from the perovskite device, electrically modulated at a given frequency, was coupled with an optical fibre (length 2 m) transmitting the optical signal. The output beam from the other end of the fibre was directed towards a high-speed silicon photodetector, whose output signals were collected by an oscilloscope. The perovskite laser shows a fast temporal response under electrical pulses (peak voltage 50 V and pulse duration 20 ns). The rise (10%–90%) and fall (90%–10%) times of the laser device were found to be 5.4 ns and 5.1 ns, respectively (Fig. 4b). The response speed of the perovskite laser is largely determined by the resistance–capacitance (RC) constant of the microcavity PeLED sub-unit[46], as its spontaneous emission lifetime is only about 2 ns under intense current injection (Supplementary Fig. 8). The transient EL intensity of the devices can be simulated by varying the RC constants (Supplementary Fig. 9 and Supplementary Note 3), consistent with the lasing intensity profiles. The perovskite laser can rapidly modulate at a bandwidth of about 36.2 MHz (Fig. 4c and Extended Data Fig. 10). The fast modulation performance was achieved by reducing the active area (from 0.02 mm$^2$ to 0.005 mm$^2$) and by using silicon substrates, which decreased parasitic capacitance and enabled device operation under intense current densities of approaching $10^3$ A cm$^{-2}$ (Extended Data Fig. 10). Stable pulse trains can be obtained from the laser at high frequencies (for example, 30 MHz) (Fig. 4d). At present, the spontaneous emission lifetimes (about 2 ns) of the microcavity PeLED sub-units limit the modulation rates of the lasers to the sub-GHz regime, in which applications, including data transmission and device interconnects, can be expected.

## Conclusion

In summary, we have demonstrated electrically driven lasing from a dual-cavity perovskite device. The device is constructed by integrating a low-threshold single-crystal perovskite microcavity with a high-power microcavity PeLED, forming a vertically stacked multi-layer structure. Under pulsed electrical excitation, the dual-cavity perovskite device

exhibits a minimum lasing threshold of 92 A cm$^{-2}$ (average threshold: 129 A cm$^{-2}$, at about 22 °C in air). This is an order of magnitude lower than the thresholds of state-of-the-art electrically driven organic lasers[9,10]. The defining characteristics of lasing, including thresholds (Fig. 3a), linewidth narrowing (Fig. 3b), polarization properties (Fig. 3c) and beam profiles (Extended Data Fig. 6) have been carefully examined according to the established protocols[43].

A distinct feature of our perovskite laser is the integrated dual-cavity device design, which allows the microcavity PeLED sub-unit to deliver concentrated optical power into the single-crystal perovskite microcavity sub-unit (with a coupling efficiency of 82.7%) to establish the lasing action. The dual-cavity architecture reduces the lasing threshold by about 4.7 times compared with a similarly prepared single-cavity device under electrical excitation. Our perovskite laser shows a $T_{50}$ lifetime of 1.8 h (6.4 × 10$^4$ voltage pulses at 10 Hz), outperforming the stability of electrically driven organic lasers[9,10]. The laser can be rapidly modulated at a bandwidth of 36.2 MHz. Our work addresses the challenge of achieving electrically driven lasing from perovskite semiconductors, creating a new class of light source for data transmission, photonic computation and biomedical applications.

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

## Methods

### Materials

Caesium iodide (CsI, 99.99%), molybdenum trioxide ($MoO_3$), silicon dioxide ($SiO_2$) and titanium dioxide ($TiO_2$) were purchased from Sigma Aldrich. Polyethylenimine (average molecular weight 50,000, 37% solution in $H_2O$) was purchased from J&K Scientific. Lead iodide ($PbI_2$, 99.99%), formamidinium bromide (FABr, 99.99%) and formamidinium iodide (FAI, 99.99%) were purchased from TCI. Sulfonamide (SFA) was purchased from Macklin Biochemical. Poly(9,9-dioctylfluorene-*co-N*-(4-butylphenyl)-diphenylamine) (TFB) and zinc oxide (ZnO) nanocrystals (10% wt in $H_2O$) were purchased from Xian Yuri Solar. Dimethyl sulfoxide (DMSO), gamma-butyrolactone (GBL), ethyl acetate, ethanol, isopropanol alcohol, *N,N*-dimethylformamide (DMF) and chlorobenzene (CB) were purchased from Aladdin. Gold pellets were purchased from Hebei Jiuyue New Material Technology. All chemicals were used as received without further purification.

### Preparation of perovskite precursor solution

The $Cs_{0.5}FA_{0.5}PbI_2Br$ precursor solution was prepared by dissolving CsI, $PbI_2$, FABr and SFA in DMF with a molar ratio of 0.5:1:0.5:0.06. The concentration of $Pb^{2+}$ in the precursor solution was 0.11 mol $l^{-1}$. For single-crystal growth, FAI and $PbI_2$ in different molar ratios were fully dissolved in a mixed solvent of GBL/DMSO (volume ratio 4:1) to obtain a 1.2 M perovskite precursor solution. The precursor solution was stirred at room temperature in an $N_2$-filled glovebox for about 2 h. The solution was filtered with polytetrafluoroethylene filters (0.22 μm) before use.

### Fabrication of DBRs

Alternating layers of silicon dioxide ($SiO_2$, $n = 1.478$) and titanium nitride ($TiO_2$, $n = 2.498$) were deposited onto glass substrates to form DBRs, using an e-beam evaporator with its chamber enclosed in a nitrogen glovebox. DBR1 in microcavity I was targeted for high reflectance in the 450–750 nm wavelength range, so that the emission at about 680 nm could be reflected and directed towards the semi-transparent Au electrode. DBR1 consists of approximately 30 pairs of $SiO_2/TiO_2$, with the thicknesses controlled to be 115 nm and 68 nm for each pair of $SiO_2$ and $TiO_2$ layers, respectively. DBR2 in microcavity II showed high reflectance (>98%) at 750–1,000 nm and high transmittance (>90%) below 680 nm. To fabricate DBR2, about 20 pairs of $SiO_2/TiO_2$ were deposited, with thicknesses controlled to be 142 nm and 85 nm for each pair of $SiO_2$ and $TiO_2$ layers, respectively. DBR3 in microcavity II may share the same design as DBR2 in unoptimized devices. In optimized devices, DBR3 is redesigned to feature high reflectance (>98%) at 620–870 nm. To fabricate the optimized DBR3, about 20 pairs of $SiO_2/TiO_2$ were deposited, with thicknesses controlled to be 125 nm and 75 nm for each pair of $SiO_2$ and $TiO_2$ layers, respectively.

### Fabrication of the single-crystal perovskite microcavity

Gold stripes (thickness about 0.2 μm) were evaporated onto the two edges of the DBR as a spacer. A pair of DBR and glass was pressed against each other and bonded together by a clip to form empty channels. A drop of $FAPbI_3$ perovskite precursor solution (5 μl) was then injected from the edge of the DBR/glass pair, penetrating the channels because of capillary force. The growth of the $FAPbI_3$ perovskite single crystals was completed in an $N_2$-filled glovebox. The annealing temperature for the samples was gradually increased from 60 °C to 130 °C at a rate of 2 °C $min^{-1}$, then kept at 130 °C for 12 h. Finally, the temperature of the samples was gradually cooled down to room temperature. The total duration of the crystal growth was about 2 days. The glass substrate was removed using a thin razor blade. The $FAPbI_3$ single crystal on DBR was then transferred into an e-beam evaporator for depositing another DBR on top of the single crystal, completing the fabrication of microcavity II. The optimum thickness of the $FAPbI_3$ single crystals was found to be about 180 nm.

### Fabrication of the integrated dual-cavity perovskite lasers

The DBR/glass substrates were sequentially cleaned using deionized water, acetone and isopropanol under ultrasonication. The substrates were dried with a nitrogen blowgun and exposed to UV–ozone for 15 min before use. ZnO nanoparticles were subjected to spin-coating onto the DBR substrates at 5,000 rpm for 45 s and then annealed at 150 °C for 10 min. Subsequently, a PEIE solution (0.04 wt% in isopropanol) was subjected to spin-coating on top of the ZnO layer at 5,000 rpm for 45 s, followed by annealing at 100 °C for 10 min. After cooling down to the ambient temperature, the samples were transferred into an $N_2$-filled glovebox. The perovskite precursor solution was subjected to spin-coating onto the PEIE-coated ZnO substrates at 5,000 rpm for 60 s, followed by annealing at 100 °C for 10 min. Next, a TFB solution (12 mg $ml^{-1}$ in chlorobenzene) was subjected to spin-coating at 4,000 rpm for 45 s. Subsequently, 7 nm of $MoO_x$ and 20 nm of Au were sequentially evaporated through a shadow mask under a high vacuum to form the semi-transparent electrode. The top surface of the resultant microcavity PeLED (microcavity I) was covered by UV epoxy (NOA 81). Following this step, microcavity II containing the $FAPbI_3$ single crystal was placed on top of microcavity I, with the single crystal positioned on top of the overlapping area of ITO and Au. The device was then exposed to UV light for 30 s to form a reliable bond between the two microcavities, completing the integration process.

### Surface morphology measurements for perovskite single crystals

The surface morphology of the perovskite single crystals was inspected using a field emission scanning electron microscope (Hitachi, SU70 SEM). The surface roughness of the samples was characterized using an atomic force microscope (Bruker, Multimode-8) under the tapping mode. The thickness of perovskite single crystals was measured using a stylus profilometer (Bruker, DektakXT).

### Scanning transmission electron microscopy experiments

A spherical aberration-corrected scanning transmission electron microscope (FEI, Titan ChemiSTEM) was used for collecting the cross-sectional images of the integrated dual-cavity laser devices. The samples for HAADF-STEM measurements were prepared using a dual-beam focused-ion-beam system (Quata 3D FEG).

### XRD measurements

The standard XRD measurements of the samples were performed using Shimadzu XRD 7000 with Cu $K\alpha_{1,2}$ radiation ($\lambda = 1.541$ Å). The measurements were performed at a continuous mode with a scan range of $10° < 2\theta < 60°$ and a scan speed of 5° $min^{-1}$. The 2D diffraction images were collected using a single-crystal X-ray diffractometer with a 2D detector (D8 Venture, Bruker). The powders scratched from the single crystals were placed on a quartz substrate. The monochromatic X-ray beam was focused onto the powders. The Bragg reflections were detected using a MAR-165 charge-coupled device detector with an exposure time of 60 s.

### Photoluminescence and optical reflection measurements

A 405-nm continuous wave laser was used to excite the $FAPbI_3$ single crystals and the $Cs_{0.5}FA_{0.5}PbI_2Br$ polycrystalline films. The photoluminescence spectra were obtained by a high-sensitivity spectrometer (QE-Pro, Ocean Optics). The reflectance of the DBRs was measured by a custom-built setup. The white light from a halogen lamp was incident normally on the DBRs through one of the ports on a Y-type fibre. The reflected light was collected through another port on the Y-type fibre and directed into a spectrometer (USB 4000, Ocean Optics). An aluminium mirror was used as the reference sample for optical reflection.

## Characterization of microcavity PeLEDs under d.c. and pulsed operations

For d.c. operation, the microcavity PeLEDs were driven by a computer-controlled source meter (Keithley 2450). The voltage was swept from 1 V to 7 V at a rate of 0.2 V per step. Simultaneously, the EL emitted from the microcavity PeLED through the semi-transparent Au electrode was measured using a calibrated photodetector (819D-SL-2-CAL, Newport) in an integrating sphere, with a silicon photodiode used in conjunction with a computer-controlled optical power meter (Newport 1936-R). The EL spectra of the PeLEDs were measured using a fibre-coupled spectrometer (QE-Pro, Ocean Optics) on another port of the integration sphere. The EQE was obtained by dividing the total number of emitted photons by the total number of injected electrons per unit time. The measurements were performed in the dark under ambient conditions.

For pulsed operation, a pulse generator (AVTECH 1010-B) was used to provide voltage pulses (pulse width 1 μs, repetition rate 100 Hz) with a d.c. offset bias of 1.7 V. The emitted optical power from the devices was measured using a silicon photodiode (FDS1010, Thorlabs). The photodiode was connected to a custom-built large-bandwidth transimpedance amplifier. The amplified signal was then sent to an oscilloscope (MDO34, Tektronix) to obtain the average optical power output based on the responsivity of the photodiode. To measure the transient profiles, the PeLED was connected in series with a 2-ohm resistor. The transient voltage signal on the resistor was amplified by a voltage amplifier (FEMTO DHPVA-101) and collected by the oscilloscope. The transient current signal was obtained by dividing the voltage by the resistance.

## Characterization of angular emission profiles of PeLEDs

To measure the angular emission profiles, the PeLED was positioned at the centre of a rotational stage, with a fibre-coupled spectrometer (QE-Pro, Ocean Optics) placed at a fixed distance. The EL spectra at different angles were measured while rotating the stage.

## Optically pumped lasing experiments

Microcavity II was pumped by 1-μs optical pulses from a modulated 405-nm continuous wave laser at a repetition rate of 10 Hz. The continuous wave laser was synchronized with an arbitrary-wave generator, which produced square waves with an amplitude of 5 V. The excitation beam was directed through a tunable neutral density filter (Thorlabs, NDC-50C-2M-B) to adjust the excitation intensity. A small fraction of the beam was directed onto a photodiode (Thorlabs, DET110) for pump power monitoring. The rest of the beam was directed into an inverted microscope and then focused onto the sample through a 50× objective, producing a spot size of around 18 μm. The output emission from the laser was measured using a fibre-coupled spectrometer (Maya 2000, Ocean Optics) with a spectral resolution of 0.08 nm. The measurements were performed under ambient conditions (temperature about 22 °C; relative humidity about 50%). For the femtosecond laser-pumped lasing, the optical excitation (400 nm; pulse duration around 270 fs; repetition rate 50 kHz) was generated using an optical parametric amplifier (Orpheus-F, Light Conversion) pumped by a 1,030 nm Yb:KGW laser (Pharos, Light Conversion).

## Characterization of electrically driven lasing

The dual-cavity perovskite device was driven by a pulse generator (AVTECH 1010-B). The output emission from the dual-cavity laser was measured using a high-resolution fibre-coupled spectrometer (Maya 2000, Ocean Optics). Different current densities were obtained by adjusting the amplitude of the voltage pulses. The far-field emission beams were captured using a high-resolution beam profiler CCD camera (Thorlabs, BC207VIS). The polarization properties of the emission were characterized based on the emission intensities measured

through a rotatable linear-polarizer plate. For the stability measurements, the laser device was driven by voltage pulses (peak current density 163 A cm$^{-2}$, pulse duration: 1 μs and repetition rate 10 Hz), with the output emission continuously measured using a fibre-coupled spectrometer (Maya 2000, Ocean Optics) with a spectral resolution of 0.08 nm. The measurements were performed under ambient conditions (temperature about 22 °C; relative humidity around 50%).

## Simulation of optical power distribution

The optical simulation was carried out using the commercial software LightTools. The optical power (irradiance) distribution at the plane of the perovskite single crystal (in microcavity II) was simulated for both single-cavity bottom-emission and dual-cavity top-emission devices. During the simulation, 2D receivers were placed at the plane of the PeLED active area (serving as the excitation source) and the perovskite single crystal within microcavity II. The PeLED active area was regarded as an assembly of numerous point-light sources and was partitioned into 1 μm × 1 μm sections. The output irradiance was normalized to a value of 100. For each point source, either a Lambertian profile or a microcavity angular profile was assigned as the angular emission profile. The optical power reaching the 2D receiver at the plane of the single-crystal layer was computed. Subsequently, all the fractional optical power was integrated for each position to obtain the overall distribution.

## Frequency-response measurements

The dual-cavity laser device was driven by a square-wave voltage signal with various frequencies, supplied by an arbitrary-wave generator. The output emission from the device was directed to a fibre-coupled avalanche photodetector (Thorlabs, APD430A2/M). The signal was fed into an oscilloscope (MDO34, Tektronix). The amplitudes of the signals were recorded at different frequencies. The 3 dB frequency or the bandwidth corresponding to the frequency at which the amplitude drops to half of its original value. The frequency-response characteristics of the integrated laser were measured similarly, but driven by a low duty-cycle (10%) pulse train.

## Data availability

The data supporting the findings of this study are available in the paper and its Supplementary Information.

**Acknowledgements** This work was supported by the National Natural Science Foundation of China (NSFC) (62405279 and 62274144), the Scientific Research Innovation Capability Support Project for Young Faculty (ZYGXQNJSKYCXNLZCXM-I25), the Natural Science Foundation of Zhejiang Province (LZ24F040002) and the Zhejiang Provincial Government. We acknowledge the technical support from the Core Facilities, State Key Laboratory of Extreme Photonics and Instrumentation, Zhejiang University. We thank M. Yu, Y. Zhao and X. Yang for their administrative support.

**Author contributions** C.Z. and D.D. planned the study and designed the experiments. C.Z. fabricated and characterized the dual-cavity perovskite lasers. Z.R. and K.H. assisted with the fabrication of the microcavity PeLED sub-unit. Z.W. and Y.F. assisted with the fabrication of the single-crystal perovskite microcavity. C.Z. carried out spectroscopic and structural experiments with the assistance of Z.W. and Y.Y.; B.Y. carried out the simulations of optical power (irradiance) distributions. C.Z. wrote the initial draft of the manuscript, which was revised by D.D. and B.Z. All authors contributed to the work and commented on the paper. D.D., C.Z. and B.Z. guided the project.

**Competing interests** D.D., C.Z. and B.Z. are inventors on CN patent application, no. 202510987102.5. The other authors declare no competing interests.

**Additional information**
**Correspondence and requests for materials** should be addressed to Chen Zou, Baodan Zhao or Dawei Di.

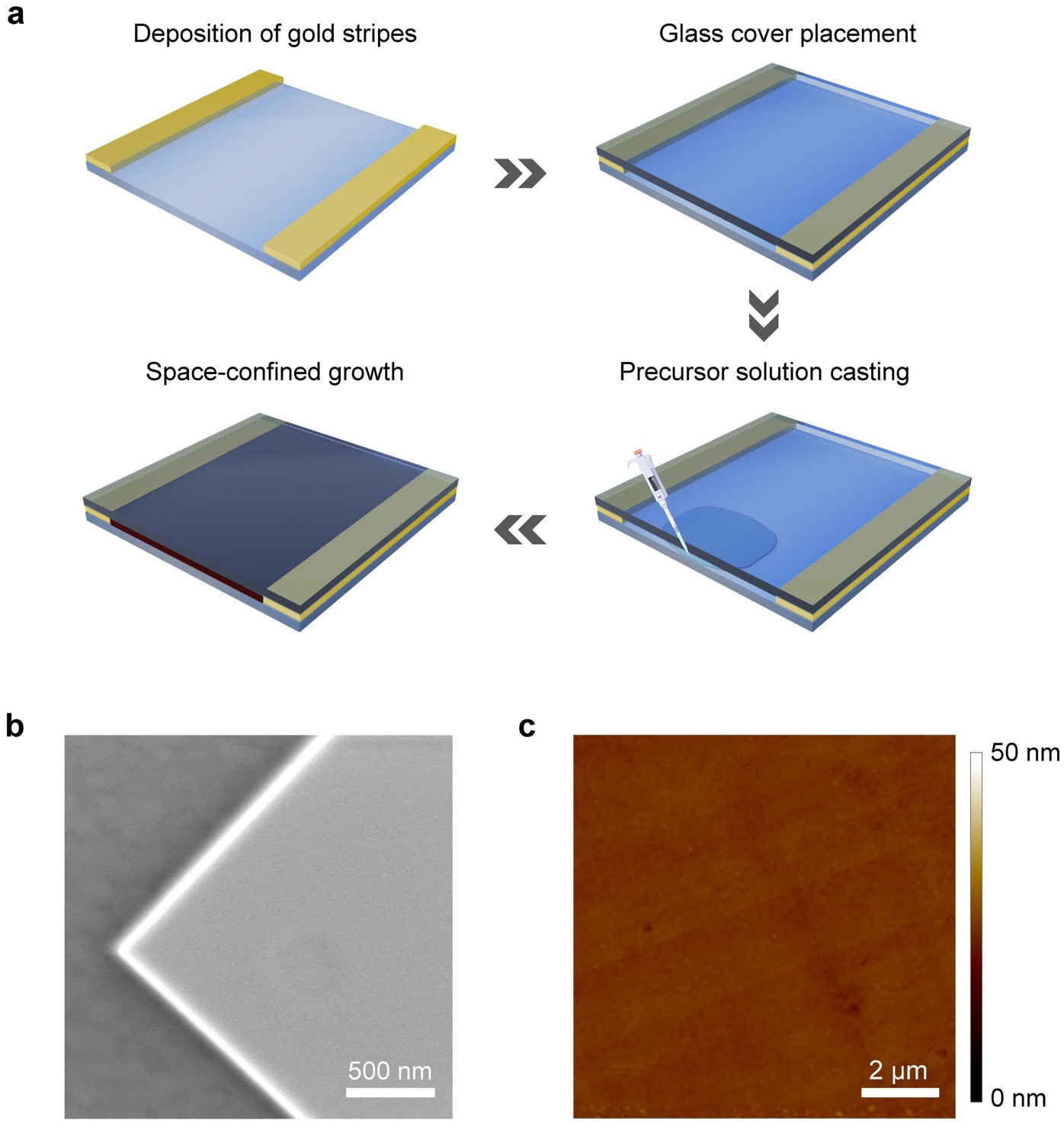

**a**

Deposition of gold stripes

Glass cover placement

Space-confined growth

Precursor solution casting

**b**

500 nm

**c**

50 nm

2 µm

0 nm

**Extended Data Fig. 1 | Space-confined growth of perovskite single crystals and their surface morphology. a**, Processes of the FAPbI$_3$ single crystal growth. **b**, Top-view SEM image of an FAPbI$_3$ single crystal on an ITO/glass substrate.

**c**, Atomic force microscope (AFM) image. The RMS roughness of the single crystal was found to be ~0.7 nm.

**a**   **b**

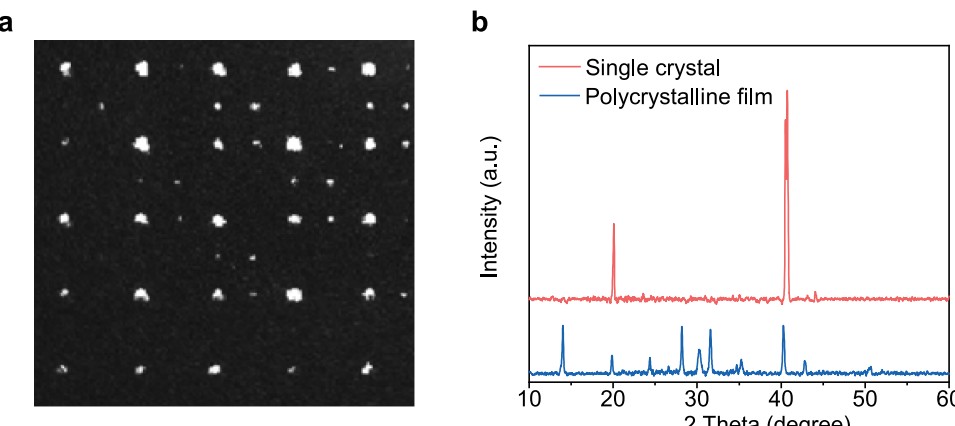

**Extended Data Fig. 2 | XRD characterization of FAPbI₃ single crystals and polycrystalline films. a**, 2D XRD image of an FAPbI₃ single crystal. **b**, XRD patterns of an FAPbI₃ single crystal and a polycrystalline FAPbI₃ film.

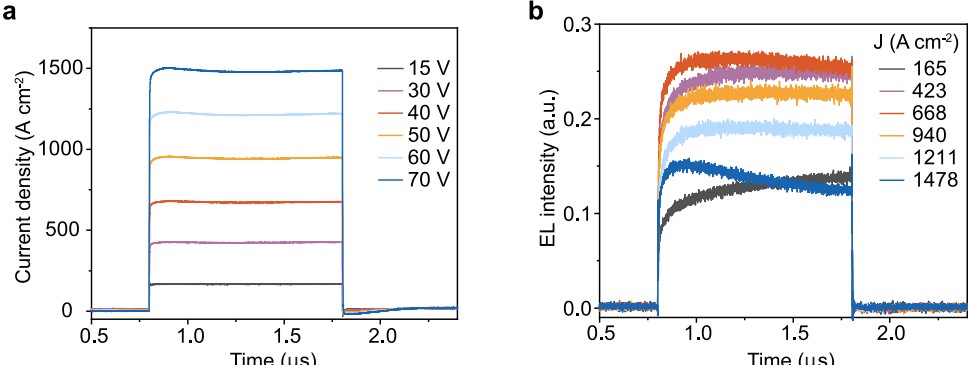

**Extended Data Fig. 3 | Transient characterization of microcavity PeLEDs under 1-µs voltage pulses. a**, Transient current density profiles under voltage pulses. **b**, Transient EL profiles under different current densities.

**a**

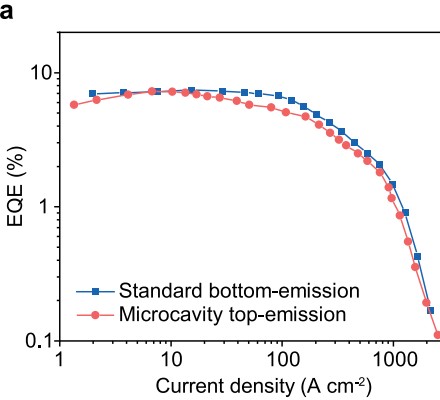
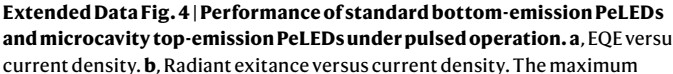

**b**

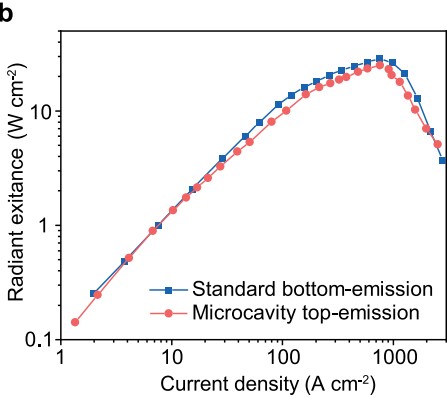

**Extended Data Fig. 4 | Performance of standard bottom-emission PeLEDs and microcavity top-emission PeLEDs under pulsed operation. a**, EQE versus current density. **b**, Radiant exitance versus current density. The maximum optical power densities are 28.5 W cm$^{-2}$ and 25.1 W cm$^{-2}$ for the standard bottom-emission and microcavity top-emission PeLEDs, respectively. The active area of the devices is 0.02 mm$^2$. Pulse duration: 1 μs, repetition rate: 100 Hz.

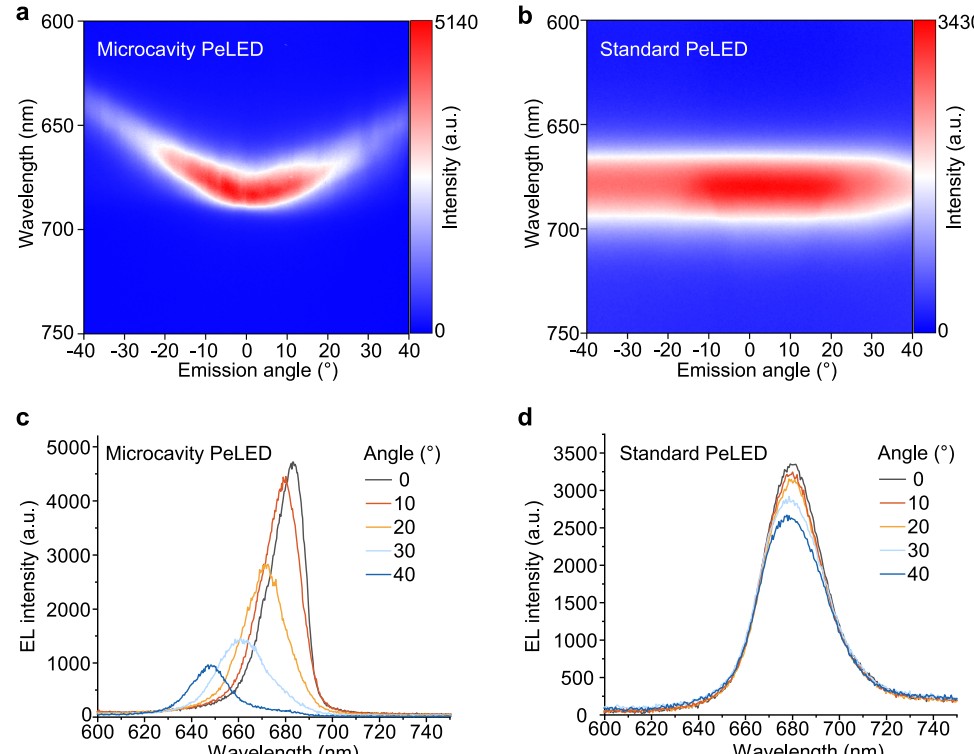

**Extended Data Fig. 5 | Angle-dependent EL of PeLEDs. a-b**, Angle-resolved EL spectra of (**a**) a microcavity PeLED and (**b**) a standard PeLED. **c-d**, EL spectra measured at 0°, 10°, 20°, 30°, and 40° from the surface normal for (**c**) a microcavity PeLED and (**d**) a standard PeLED.

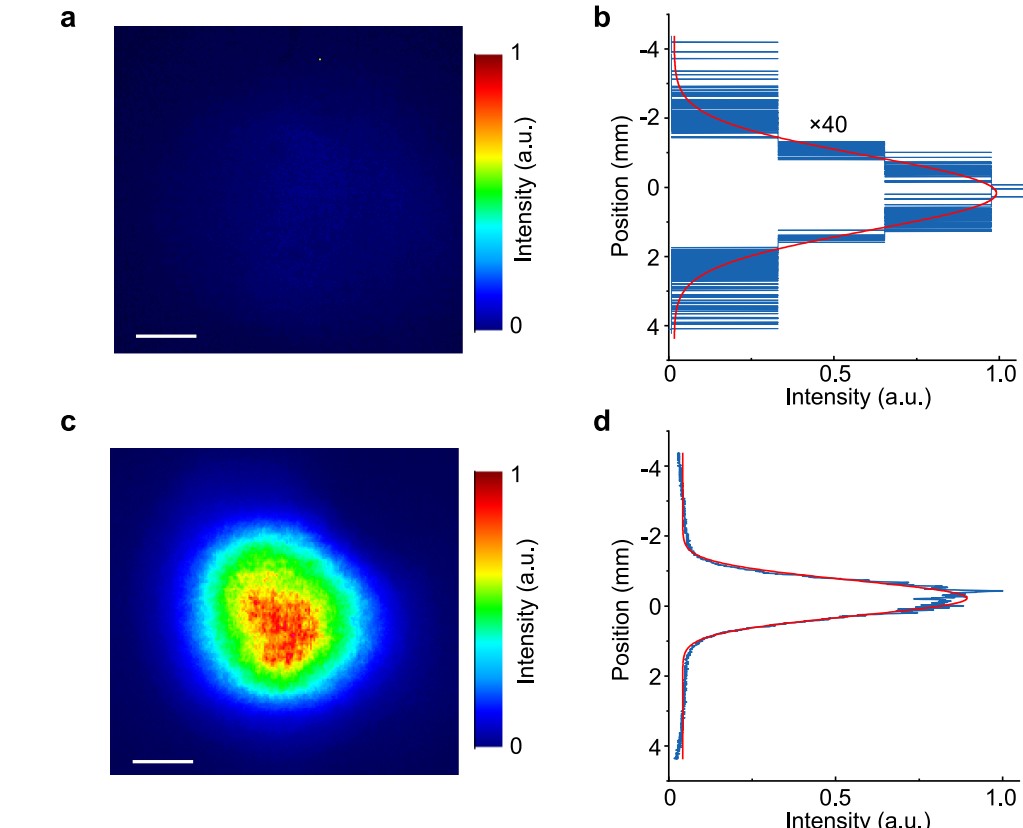

**Extended Data Fig. 6 | Far-field beam profiles and line-intensity profiles of electrically-driven perovskite lasers, measured at 5 cm distance from the device. a**, Far-field beam profile and **b**, the corresponding line-intensity profile (blue curve, magnified by 40 times) measured at 85 A cm$^{-2}$ (below threshold) (the red curve represents the gaussian fit). **c**, Far-field beam profile and **d**, the corresponding line-intensity profile (blue curve) measured at 280 A cm$^{-2}$ (above threshold) (the red curve represents the gaussian fit). The scale bars in **a** and **c**: 1 mm. The measurements were performed at room temperature in air (temperature: ~22 °C, relative humidity: ~50%). The FHWMs of the beam profiles below and above the threshold are 2.52 mm and 1.18 mm, respectively.

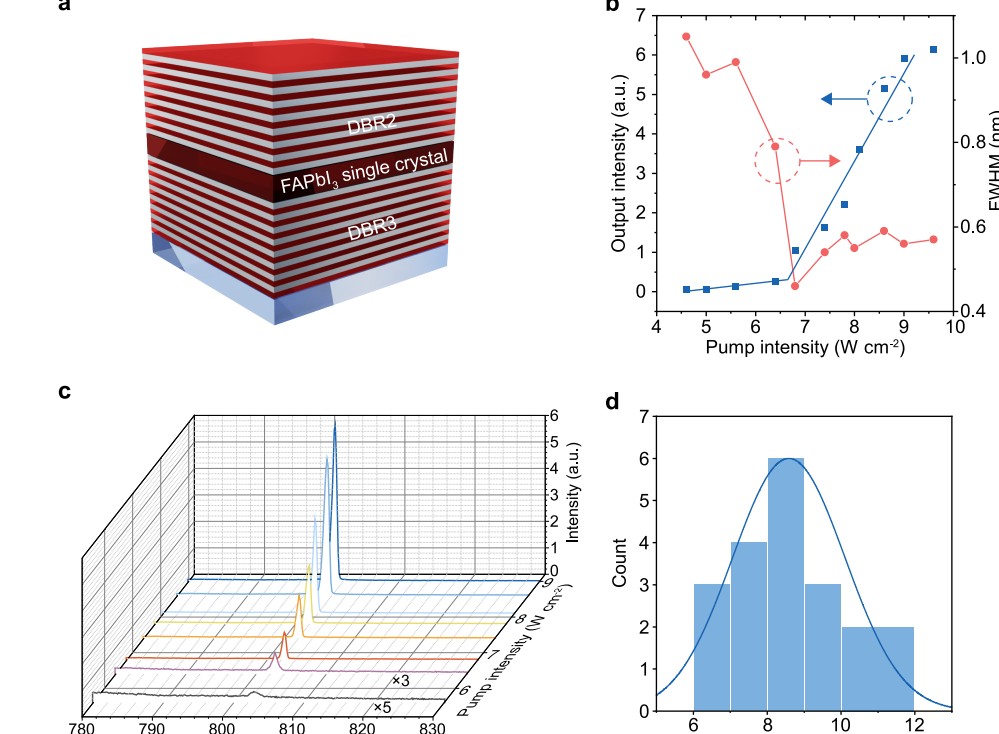

**Extended Data Fig. 7 | Characterization of single-crystal perovskite microcavities under optical pumping. a**, Structure of a single-crystal perovskite microcavity. **b**, Output optical power intensity and FWHM versus pump intensity, showing clear transitions around the lasing threshold of ~6.6 W cm$^{-2}$. **c**, Emission spectra under different pump intensities. The intensities of the emission spectra below the threshold are magnified by the factors specified near the curves. **d**, Histogram of the lasing thresholds. The measurements were performed at room temperature in air (temperature: ~22 °C, relative humidity: ~50%). The single-crystal perovskite microcavities were pumped by 1-μs optical pulses from an electrically modulated 405-nm continuous-wave (CW) laser at a repetition rate of 10 Hz. The diameter of the focused pump spot was ~18 μm.

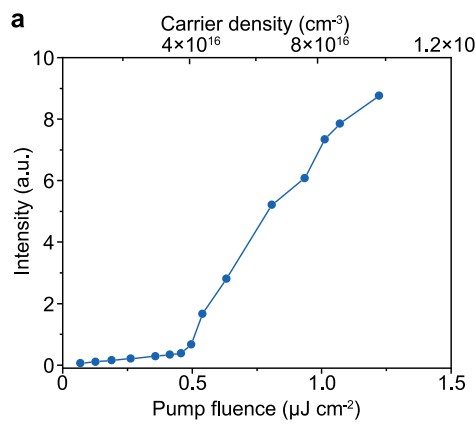

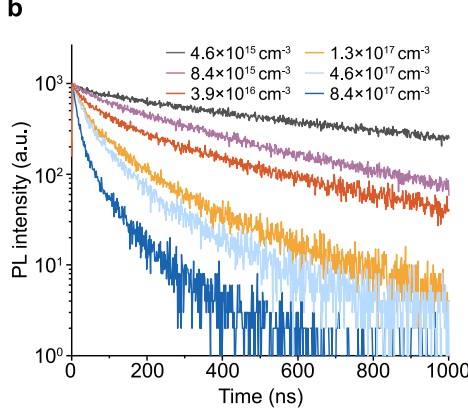

**Extended Data Fig. 8 | Emission intensity of microcavity II under pulsed optical pumping, and PL decay kinetics of a perovskite single crystal.** **a**, Output intensity of microcavity II versus pump fluence under femtosecond optical pumping (400 nm; pulse duration: ~270 fs; repetition rate: 50 kHz). **b**, PL decay kinetics of a perovskite single crystal at different carrier densities.

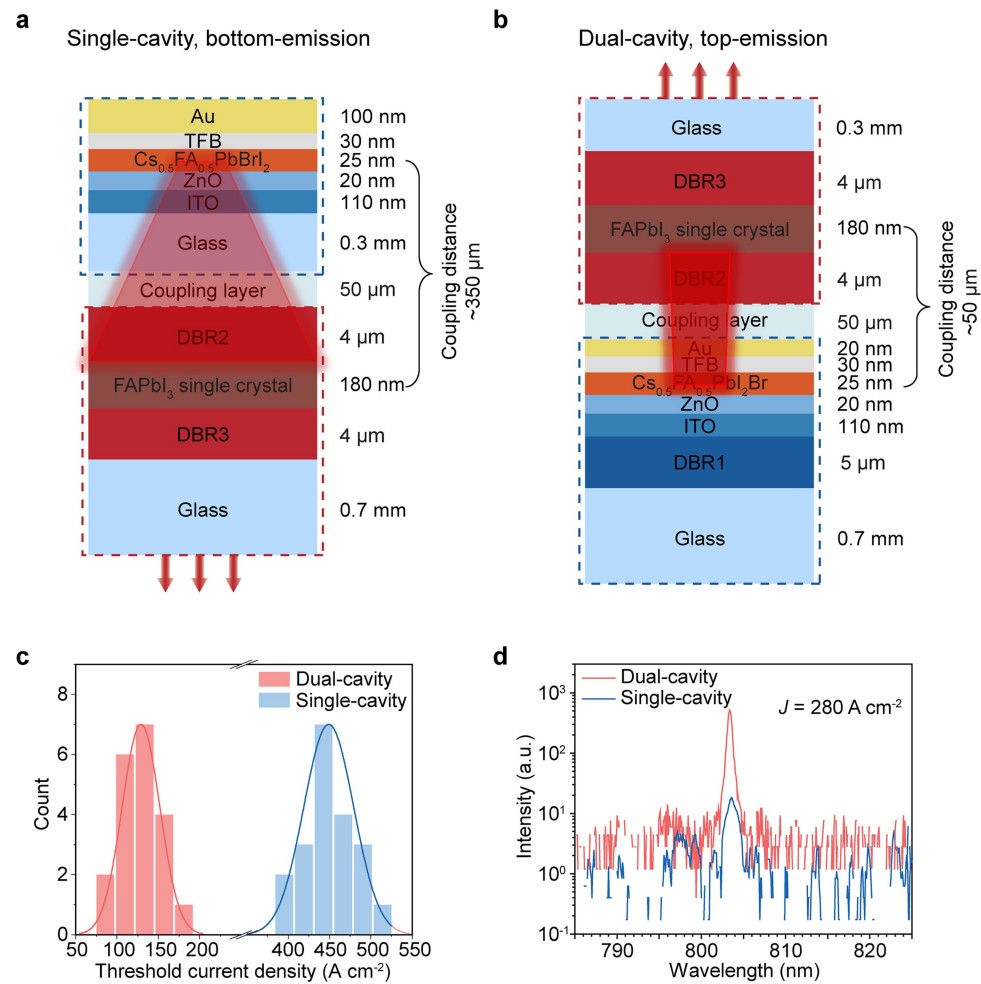

**Extended Data Fig. 9 | Comparison of electrically-driven integrated perovskite lasers with single-cavity and dual-cavity structures. a**, Integrated perovskite laser based on a single-cavity structure. **b**, Integrated perovskite laser based on the dual-cavity structure. **c**, Histogram of current-density thresholds for the two types of perovskite lasers. **d**, Emission spectra of the perovskite lasers driven at a current density of 280 A cm⁻². The measurements were performed at room temperature in air (temperature: ~22 °C, relative humidity: ~50%).

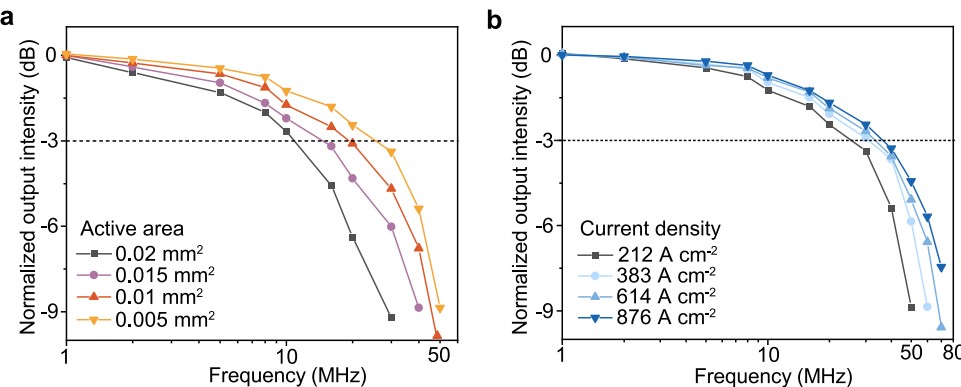

**Extended Data Fig. 10 | Frequency response of the electrically-driven dual-cavity perovskite lasers. a**, Frequency response characteristics of dual-cavity lasers with different active areas. **b**, Frequency response characteristics of a dual-cavity laser (active area: 0.005 mm²) at different current densities.