## [Peer Review file · Nature]

Electrically-driven lasing from a dual-cavity perovskite device

Corresponding Author: Professor Dawei Di

Version 0:

Reviewer comments:

Referee #1

(Remarks to the Author)

The manuscript from D. Di et al. reports an electrically driven perovskite laser realized through an ingenious double cavity architecture, where a high-power perovskite LED integrated in a microcavity is coupled to a second low-threshold, single-crystal microcavity cavity stacked on top of the first one. The work is novel and of high quality, with results that could establish a milestone in the progress of perovskite-based lasing devices.

However several points need clarification before the manuscript is considered for publication. In particular the achievement of electrically-driven laser is due to the exceptional performances of the two sub-unities of the device, the high-power LED cavity and the low-threshold single-crystal cavity. In order to provide true value for the community, such performances need to be explained on a rational base. The recommendation is therefore that the manuscript is considered for publication contingent on the authors being able to address the following criticism.

- Concerning the low-threshold, single-crystal cavity, the stimulated emission threshold in cw is less than 10 W/cm^2 under optical pumping. In order to reach the population inversion and produce the same carrier concentration observed under pulsed optical excitation (at least 10^{19} cm^{-2} in perovskites, obtained with around 10 microJ/cm^2 ultrafast pulses), the reported threshold would require a carrier lifetime around 1 microsecond, at least two orders of magnitude longer than what is observed under intense excitation. Therefore the authors should clarify if this is indeed the case or if, more probably, stimulated emission is not provided by intrinsic states, but by defect states. In case defect states are responsible, how reproducible is the threshold? Under what conditions are defect obtained?

- Polarization is employed as a signature for stimulated emission. It is however unclear what element in the whole structure is polarization sensitive and why a peculiar polarization direction is preferred. The polarization is the same in all instances and in all devices?

- Regarding the high-power LED cavity, since the active area is such a crucial parameter (Fig. 2a), the authors should clarify if there are current funneling effects through a narrow channel that locally increase the current and/or optical emission density.

- The authors should declare the thickness and the size of the crystal, that would be crucial to replicate the experiment and assess the excitation density.

- In page 4 line 97, it is mentioned that the area of the microcavity is reduced from 5.25 mm^2 to 0.02 mm^2 , it is not clear to me what is the reason for these values, the sentence should be probably rephrased.

Comments on Lasing reporting summary:

Point 2: linewidth narrowing. I suggest to add the spectral resolution of the measurements of Electrically driven lasing (page 12 line 363), if it is the same of Optically pumped lasing it should be mentioned.

Point 7: theoretical analysis. Simulations are used only to evaluate the inter-cavity coupling efficiency, but are not used extensively to simulate the whole emission process. More detailed simulations could perhaps be used to understand the rise

and fall times reported in page 7 line 190 and provide indications on if and how the device could be improved towards higher modulation bandwidths.

Point 9: There is an inconsistency about the value of the lasing threshold of the dual cavity system, that in the abstract is reported to be $92\text{W}/\text{cm}^2$ while in the Lasing Reporting Summary is $129\text{A}/\text{cm}^2$ (Point 9).

Other comments:

- The manuscript could improve by stating clearly from the beginning that the single crystal microcavity is optically pumped by the PeLED. I suggest to move the sentence "The EL from the microcavity PeLED (microcavity I) was absorbed by the FAPbI₃ single crystal whose emission was the amplified in the second microcavity (microcavity II)." from Methods section "Fabrication of the DBRs" (page 9, lines 251- 253) to somewhere in page 2 in the main text.

- Given the broad readership of Nature, authors could consider to add to Fig. 1 a sketch describing the working mechanism of the device (similarly to Fig. 4 A).

Page 2, line 65: crystalization -> crystallization

Page 11, line 305: x-ray -> X-ray

Page 11, line 322: integration sphere -> integrating sphere

Referee #2

(Remarks to the Author)

The authors present a perovskite laser device consisting of vertically integrated dual-cavity structure. One of the cavities is a light-emitting diode, which provides optical power into the other cavity (VCSEL). In contrast to previous studies proposing the similar device architectures, i.e. vertically-integrated LED/VCSEL [ref. 10 and a very recent paper reported from the authors' group (<https://doi.org/10.1126/sciadv.adr8826>)], the current study demonstrates more efficient power delivery owing to the cavity structure enabling the directional optical output. As a consequence, the authors achieve a current-injected perovskite laser module for the first time. The quality of data and presentation is extremely high. Experimental results are presented with appropriate use of statistics and uncertainties are removed.

The novelty of this study, the authors claim, lies in the dual cavity structure. This is certainly true. Furthermore, the authors' group is one of the world's leading groups in perovskite LEDs, and the high-power LEDs presented in this study are likely the result of the authors' maximum use of their technology. However, this study can be viewed as nothing more than a combination of existing technologies, LED microcavity (e.g. <https://doi.org/10.1063/1.116858>) and the integrated LED/VCSEL architecture (ref. 10), and in my opinion, academic progress of this study, as it stands, is insufficient for Nature. Specifically, the authors demonstrate an electrical modulation of their device, but unfortunately, this may expose weaknesses in the authors' device architecture. This is because the modulation rate may be limited by the spontaneous emission lifetime that is the intrinsic operation mechanism for LEDs. This is in contrast to the real LDs (> ~GHz), which utilize the stimulated emission. If the authors can address this concern and demonstrate that their device is superior in terms of modulation speed (or otherwise), it would provide one important scientific advance.

Other minor points raised are as follows.

(Definition of 'cavity')

- Especially for Microcavity 1, experimental data demonstrating that the optical output is the cavity mode are missing. Does the output from Microcavity 1 before integration with Microcavity 2 show angular dependence of the cavity mode?

- I understand that the top reflector of Microcavity 1 is the semi-transparent Au. If so, the authors need to show that the Au film exhibits sufficient reflectivity as a VCSEL.

- What is the physical cavity length in Microcavity 1? In general, perovskite LEDs are composed of emissive and carrier injection layers with a very small thickness (< ~100 nm). If the cavity length is too small, a cutoff condition is imposed on the standing waves in the microcavity. The authors have to exclude this concern.

- Just the confirmation, Are DBR 2 and 3 based on the same design?

(Hyperbole)

- In line 104 (page 4), the term "the highest power output" is misleading as the authors' LED has a footprint as small as 0.02 mm² the result is in the form of radiance. It is desirable to accurately claim that power can be delivered with high density.

- In Fig. 3e (page 6), It is not fair to compare performance with current injected devices (ref. 9) as the physics that gives the operating principle is very different from the authors' device.

- The authors have very recently published a similar device structure. That study needs to be cited (<https://doi.org/10.1126/sciadv.adr8826>).

Referee #3

(Remarks to the Author)

I co-reviewed this manuscript with one of the reviewers who provided the listed reports.

Referee #4

(Remarks to the Author)

Zou et al. report an electrically-driven halide perovskite laser diode via the same indirect pumping scheme recently used to realize organic laser diodes. The devices implement a DBR microcavity with a single crystal FAPbI₃ gain medium on top of a metal-DBR CsFAPbBrI perovskite LED and show a clear intensity threshold, linewidth narrowing, spontaneous polarization, and increased directionality at threshold.

With appropriate clarification in regard to the questions below, I believe the authors have successfully demonstrated lasing from their devices. Though not particularly novel from a fundamental standpoint given the similarity to the approach taken in Ref. 10, this is certainly an important step forward for the perovskite LED/lasing field and is likely to stimulate more work on indirectly-pumped perovskite laser diode devices in the future. From this standpoint, I think this manuscript merits consideration by Nature, though the following questions/suggestions should be addressed before any decision can be made:

1. The authors should clearly show the below threshold spectra in Fig. 3a so that readers can clearly see the spectral collapse for themselves. Simply magnifying the spectra vertically and indicating the magnification factor, e.g. 'x50' or similar designation, would address this. Same concern for the spectra in Ext. Data Fig. 6.
2. Given that this is a VCSEL, what breaks the polarization degeneracy above threshold and leads to quasi-linear polarization in Fig. 3c? On a related note, why is the laser emission only weakly polarized? Are there multiple transverse modes lasing, or are there different areas of the DBR/single crystal cavity that are lasing simultaneously? Do the authors observe the same polarization behavior under optical pumping? One way to address this would be to tightly focus the optical pump spot (e.g. $\sim 10 \mu\text{m}$ diameter) and scan it laterally to see whether the polarization changes locally. As it stands, the spot size of the optical pumping beam in Ext. Fig. 6 should be stated.
3. The authors attribute the lower threshold of their dual cavity device relative to the single cavity to the enhanced directionality of the LED in the former. However, my guess is that the lower threshold has much less to do with the small increase in directionality from Fig. 2f and much more to do with the higher coupling distance due to the $300 \mu\text{m}$ thick glass substrate in the latter that effectively results in a $350 \mu\text{m}$ thick coupling layer (this should be more clearly indicated in Ext. Fig. 7a), which provides more distance for the light to disperse (e.g. the coupling distance is nearly twice the device dimension). The authors should provide an accurate account of how much the increased coupling distance vs. change in LED directionality actually contributes to the lower threshold of the dual cavity device.
4. What is the distance between the laser and the detector in the far-field beam profile measurement in Ext. Fig. 5? Again, please adjust the scaling so that readers can tell for themselves what the FWHM of the curve in panel (b) is.
5. Improving the English grammar throughout would help with readability.

Version 1:

Reviewer comments:

Referee #1

(Remarks to the Author)

The manuscript 2025-01-02269A by Zou and coworkers has been thoroughly revised. The authors have seriously considered the criticisms in the original referee reports and have addressed them with scientifically sound arguments, data and analyses.

It is my opinion that the revised manuscript represents a significant improvement and overcomes the threshold for publication.

Referee #2

(Remarks to the Author)

The authors present a perovskite laser device consisting of vertically integrated dual-cavity structure. They responded appropriately to the recent reviewers' comments and revised their paper almost sufficiently. Specifically, the authors claimed, in response to my earlier comment, that the novelty of their study was that current-injected laser operation was achieved at low threshold. This is certainly true and would be a milestone toward a true current-injected laser device, removing my earlier concerns about the novelty.

It is also commendable that additional experiments on the modulation operation demonstrated that the RC time constant of the LED device is the main obstacle limiting the modulation speed. However, even if the RC issue is completely removed in the future, GHz modulation cannot be achieved as long as the PL lifetime is on the order of 2ns. Because of such physical facts, I have concerns about the authors' conclusion that their dual-cavity perovskite laser has potential for communication applications (the final part of page 8). In fact, tens of MHz would be sufficient for spatial visible light communications, in which case LEDs are used instead of LDs.

Could the authors link their modulation experiments to other applications? Or, could the authors show the potential for modulation operation of their high-power LEDs in the GHz range? I believe this point is an important one that increases the liability value of this paper.

Referee #3

(Remarks to the Author)

I co-reviewed this manuscript with one of the reviewers who provided the listed reports.

Referee #4

(Remarks to the Author)

The authors have addressed my concerns.

Response to Review Comments:

Referee #1

The manuscript from D. Di et al. reports an electrically driven perovskite laser realized through an ingenious double cavity architecture, where a high-power perovskite LED integrated in a microcavity is coupled to a second low-threshold, single-crystal microcavity cavity stacked on top of the first one. The work is novel and of high quality, with results that could establish a milestone in the progress of perovskite-based lasing devices.

However several points need clarification before the manuscript is considered for publication. In particular the achievement of electrically-driven laser is due to the exceptional performances of the two sub-unities of the device, the high-power LED cavity and the low-threshold single-crystal cavity. In order to provide true value for the community, such performances need to be explained on a rational base. The recommendation is therefore that the manuscript is considered for publication contingent on the authors being able to address the following criticism.

Response: We are grateful to the reviewer for the supportive and encouraging comments, which have guided us to strengthen our paper substantially.

- Concerning the low-threshold, single-crystal cavity, the stimulated emission threshold in cw is less than 10 W/cm^2 under optical pumping. In order to reach the population inversion and produce the same carrier concentration observed under pulsed optical excitation (at least 10^{19} cm^{-2} in perovskites, obtained with around 10 microJ/cm^2 ultrafast pulses), the reported threshold would require a carrier lifetime around 1 microsecond, at least two orders of magnitude longer than what is observed under intense excitation. Therefore the authors should clarify if this is indeed the case or if, more probably, stimulated emission is not provided by intrinsic states, but by defect states. In case defect states are responsible, how reproducible is the threshold? Under what conditions are defect obtained?

Response: We thank the reviewer for the constructive comments. Following the reviewer's advice, we analyse below whether the excitation densities at the thresholds are sufficiently high to support lasing.

From our pulsed optical pumping experiments, we estimate the excitation density (n) in the single-crystal perovskite microcavity sub-unit (microcavity II) based on the following equation (*Light Sci. Appl.* 11, 8, 2022):

$$n = \frac{\alpha P}{h\nu d}$$

where P is the excitation fluence, $h\nu$ is the excitation photon energy (3.1 eV), d is the thickness of the perovskite single crystal ($\sim 180 \text{ nm}$), and α is the optical absorption ($\sim 75\%$) obtained by

considering the measured reflection and transmission of the sample at the excitation wavelength. The lasing threshold of microcavity II is found to be $0.47 \mu\text{J cm}^{-2}$ under femtosecond laser pumping (Extended Data Fig. 8a). According to the equation above, the excitation density at the threshold is estimated to be $3.9 \times 10^{16} \text{ cm}^{-3}$ ($7.0 \times 10^{11} \text{ cm}^{-2}$). Similarly, a threshold of $10 \mu\text{J cm}^{-2}$ assumed by the reviewer corresponds to an excitation density of $8.2 \times 10^{17} \text{ cm}^{-3}$ ($1.6 \times 10^{13} \text{ cm}^{-2}$). The PL decay kinetics of the perovskite single crystals were measured under different excitation densities (Extended Data Fig. 8b). The carrier lifetimes were found to be $\sim 134.7 \text{ ns}$ at the excitation density ($3.9 \times 10^{16} \text{ cm}^{-3}$, as estimated above) equivalent to the lasing threshold ($0.47 \mu\text{J cm}^{-2}$).

Next, we estimate the CW threshold of optically pumped lasing from microcavity II through the following equation (*Nat. Commun.* 10, 988, 2019):

$$P_{th,CW} \cdot \tau = F_{th}$$

where F_{th} is the pulsed threshold ($0.47 \mu\text{J cm}^{-2}$), $P_{th,CW}$ is the CW threshold (W cm^{-2}), and τ is the carrier lifetime, which is $\sim 134.7 \text{ ns}$ at the excitation density equivalent to the lasing threshold ($0.47 \mu\text{J cm}^{-2}$). Using this equation, the CW threshold ($P_{th,CW}$) is estimated to be 3.4 W cm^{-2} , in reasonable agreement with the experimentally measured CW threshold (6.6 W cm^{-2}) (Extended Data Fig. 7). In summary, the low CW lasing thresholds are consistent with the low pulsed thresholds and the long carrier lifetimes near the threshold. The low thresholds in turn ensure low excitation densities, leading to longer lifetimes at just below the thresholds.

Therefore, the lasing thresholds can be justified without having to assume the contribution from defect states. Further, we confirm that the lasing thresholds we measured (both optically and electrically) are reproducible (Extended Data Fig. 7d and 9c).

To address this point, new experiments and analysis of the lasing thresholds as well as the histograms showing reproducibility are presented in Extended Data Fig. 8 and Extended Data Figs. 7d & 9c, with discussions provided on page 7 (highlighted) and Supplementary Note 2 of the revised manuscript.

Extended Data Fig. 8 | Emission intensity of microcavity II under pulsed optical pumping, and PL decay kinetics of a perovskite single crystal. a, Output intensity of microcavity II versus pump fluence under

femtosecond optical pumping (400 nm; pulse duration: ~ 270 fs; repetition rate: 50 kHz). **b**, PL decay kinetics of perovskite single crystals at different carrier densities.

Extended Data Fig. 7 | **d**, Histogram of the lasing thresholds under optical pumping.

Extended Data Fig. 9 | **c**, Histogram of current-density thresholds for the two types of perovskite lasers.

- Polarization is employed as a signature for stimulated emission. It is however unclear what element in the whole structure is polarization sensitive and why a peculiar polarization direction is preferred. The polarization is the same in all instances and in all devices?

Response: We thank the reviewer for raising this important point. In a planar VCSEL-type device structure with perfect axial symmetry, the lasing output should not exhibit preferential orientations of linear polarization (*ACS Photonics* 4, 2486-2494, 2017). However, it is difficult in practice for such a device to achieve ideal symmetry and isotropy. Structural imperfections, defects and anisotropy are usually present, leading to some preferred directions of polarization. Such polarization characteristics were observed in VCSELs based on a wide variety of gain media, including III-V semiconductors (*Nature* 443, 409-414, 2006; *Opt. Express* 24, 15546-

15553, 2016), perovskites (*Adv. Mater.* 29, 1604781, 2017; *Nanophotonics*, 12, 2133-2143, 2023; *Opt. Lett.* 50, 702-705, 2025; *Adv. Opt. Mater.* 9, 2001982, 2021; *Appl. Phys. Lett.* 120, 121104, 2022; *ACS Photonics* 4, 2486–2494, 2017), organics (*Adv. Funct. Mater.* 30, 2003294, 2020), and transition-metal dichalcogenides (*Nano Lett.* 21, 3331-3339, 2021).

In our perovskite devices, linearly polarized lasing output was observed, possibly owing to minor thickness variation, anisotropic photonic disorders, formation of strain during crystal growth and DBR deposition, and asymmetric crystal structures (*ACS Photonics* 4, 2486-2494, 2017; *Adv. Mater.* 29, 1604781, 2017; *Nature* 443, 409-414, 2006; *Appl. Phys. Lett.* 77, 1590-1592, 2000; *Opt. Lett.* 50, 702-705, 2025). The conformal coating process could transfer the surface roughness from the underlying perovskite crystals to the DBRs, resulting in structural non-uniformity across multiple interfaces (*Adv. Mater.* 29, 1604781, 2017; *ACS Photonics* 4, 2486-2494, 2017). Concurrently, the residual stress generated during the crystal growth and DBR deposition may introduce anisotropic strain distribution that breaks the inherent optical isotropy of the system (*Applied Physics Letters* 77, 1590-1592, 2000). The combined effect of structural asymmetry and material anisotropy is expected to create a preferential polarization direction for photon emission (Fig. 3c). As a result, the orientations of linear polarization of the electrically driven lasing from our dual-cavity perovskite devices exhibit device-to-device variation, as shown in Supplementary Fig. 6.

To address this issue, Supplementary Fig. 6 has been provided, with a clarification on the polarization of lasing on page 7 (highlighted) of the revised manuscript.

Supplementary Fig. 6 | Polarization plots of dual-cavity perovskite lasers, showing device-to-device variations in the orientations of polarization. The devices were driven at a current density of $\sim 280 \text{ A cm}^{-2}$ (above threshold).

- Regarding the high-power LED cavity, since the active area is such a crucial parameter (Fig. 2a), the authors should clarify if there are current funneling effects through a narrow channel

that locally increase the current and/or optical emission density.

Response: We are grateful for the insightful comments. The active area of each device is defined by the overlapping region of the ITO and Au electrodes (Fig. 1d). The reasonably uniform EL intensity profile of the active area (Supplementary Fig. 4) indicates the limited influence of current funneling effects arising from narrow current pathways.

To address this issue, the EL intensity profile of a microcavity PeLED is shown in Supplementary Fig. 4, with relevant discussions provided on page 4 (highlighted) of the revised manuscript.

Supplementary Fig. 4 | EL intensity profile of a microcavity PeLED. a, Optical microscopy image (dark-field) of a working microcavity PeLED (active area, 0.02 mm²). **b,** EL intensity profile extracted along the blue dashed line in **a**.

- The authors should declare the thickness and the size of the crystal, that would be crucial to replicate the experiment and assess the excitation density.

Response: We appreciate the helpful suggestion. The thickness of our perovskite single crystals is ~180 nm, as obtained from the cross-sectional SEM image in Fig. 1b. The typical size of the crystals is about 0.08 mm² (0.29 mm × 0.27 mm), as exemplified by the optical microscope image in Supplementary Fig. 1).

To address this issue, we have provided further details about the perovskite single crystals on page 2 (highlighted) of the revised manuscript.

- In page 4 line 97, it is mentioned that the area of the microcavity is reduced from 5.25mm² to 0.02mm², it is not clear to me what is the reason for these values, the sentence should be probably rephrased.

Response: We thank the reviewer for the comments. The active area of each device is defined by the overlapping region of the ITO and Au electrodes, as shown in Fig. 1d. The largest active area of 5.25 mm² corresponds to the area of a rectangle with dimensions of 3.5 mm (width of

ITO) \times 1.5 mm (width of Au); these are the specifications of the standard-sized perovskite LEDs in our lab. Narrower ITO and Au stripes were used to create smaller active areas. For example, the smallest active area of 0.02 mm² was formed by a square with dimensions of 0.14 mm (width of ITO) \times 0.14 mm (width of Au).

To clarify this point, we have provided more detailed information on the active areas of the devices in the caption of Fig. 2c (highlighted) of the revised manuscript.

Comments on Lasing reporting summary:

Point 2: linewidth narrowing. I suggest to add the spectral resolution of the measurements of Electrically driven lasing (page 12 line 363), if it is the same of Optically pumped lasing it should be mentioned.

Response: We appreciate the valuable comments. We confirm that the spectral resolution of the measurements for the electrically driven lasing is the same as that of the optically pumped lasing.

To address this point, we have now clarified the spectral resolution of the measurements in the Methods section under “Characterization of electrically-driven lasing”, on page 13 of the revised manuscript. We have also updated the lasing reporting checklist to make clear that measurements on both types of lasing have the same spectral resolution as specified.

Point 7: theoretical analysis. Simulations are used only to evaluate the inter-cavity coupling efficiency, but are not used extensively to simulate the whole emission process. More detailed simulations could perhaps be used to understand the rise and fall times reported in page 7 line 190 and provide indications on if and how the device could be improved towards higher modulation bandwidths.

Response: We appreciate the constructive comments from the reviewer. At above the lasing threshold, the carrier lifetimes in the single-crystal perovskite microcavity sub-unit (microcavity II) are on the order of picoseconds, as this is a typical carrier lifetime for perovskite lasers (*Chem. Eng. J.* 420, 127660, 2021; *Nanophotonics*, 12, 2133-2143, 2023; *Opt. Lett.* 50, 702-705, 2025). The overall rise and fall times of the device are on the order of tens of nanoseconds, and are therefore limited by the transient characteristics of the microcavity PeLED sub-unit (microcavity I), whose response time is the sum of the effective carrier lifetime (τ_{carrier}) and the resistance-capacitance (RC) time constant (τ_{RC}), where τ_{RC} is the product of the differential resistance and the parasitic capacitance of the device ($\tau_{\text{RC}} = R \cdot C$). The total response time of the device can be written as

$$\tau_{\text{response}} = \tau_{\text{carrier}} + \tau_{\text{RC}}$$

Under intense current injection ($>200 \text{ A cm}^{-2}$, above the lasing threshold), the measured rise time is $\sim 24.4 \text{ ns}$ for the microcavity PeLED sub-unit with an active area of 0.02 mm² (Supplementary Fig. 9b), which is likely dominated by the relatively large RC constant (τ_{RC})

of the device, considering the spontaneous lifetime of the microcavity PeLED is less than 2 ns under intense current injection (Supplementary Fig. 8). The differential resistance (R) can be obtained by $R=dV/dI$ (Supplementary Fig. 9a), which is estimated to be 243Ω at a current density of 212 A cm^{-2} . Further, we calculate the capacitance (C) using

$$C = \frac{\epsilon_r S}{4\pi k d}$$

where ϵ_r is the relative permittivity of perovskite ($\epsilon_r=32$) (*Nat. Photon.* 12, 681-687, 2018), S is the facing area of the two plates (0.02 mm^2); d is the distance between the two plates (90 nm), and k is the Coulomb's constant ($9.0 \times 10^9 \text{ N m}^2/\text{C}^2$). Using this equation, the capacitance is calculated to be 62.9 pF. The RC constant can be found using $\tau_{RC} = R \cdot C = 15.2 \text{ ns}$. The transient EL intensity (L) from the microcavity PeLED sub-unit during the rise stage can be described by (*J. Phys. Chem. Lett.* 14, 1777-1783, 2023)

$$L = \tanh^2\left(\frac{t}{\tau_{response}}\right)$$

where $\tau_{response} \approx \tau_{RC}$, and t is the time after the onset of electrical trigger. Using this equation, we have simulated the transient EL response of the microcavity PeLED (Supplementary Fig. 9b). For a τ_{RC} of 15.2 ns, the simulated rise time of the device is 23 ns, in good agreement with the experimentally measured rise time (24.4 ns) (Supplementary Fig. 9b). The fall time of the EL intensity shows a similar dependence on the RC constants.

Supplementary Fig. 8 | Transient PL decay curves at different current densities for a microcavity PeLED with an active area of 0.02 mm^2 . The microcavity PeLED was excited by a femtosecond laser (excitation wavelength: 400 nm; pulse duration: $\sim 270 \text{ fs}$, excitation fluence: $\sim 1 \mu\text{J cm}^{-2}$), while driven at constant current densities of 5 A cm^{-2} , 8 A cm^{-2} and 10 A cm^{-2} . The spontaneous emission lifetime of the microcavity PeLED is about 2 ns.

Supplementary Fig. 9 | Differential resistance and transient EL of the microcavity PeLED sub-unit. a, Differential resistance as a function of current density. **b,** Simulated transient EL of microcavity PeLEDs with different RC time constants (using Equation S3-3 in Supplementary Note 3). An RC constant of 15.2 ns (red curve) yields a simulated EL response with a rise time of 23 ns, which closely agrees with the experimental data (rise time: 24.4 ns). Active area of the device: 0.02 mm². Current density: ~212 A cm⁻².

Inspired by the reviewer's comments, we have carried out further experiments to improve the modulation bandwidth of the perovskite lasers. During the revision process, the bandwidth was improved by reducing the RC constant and increasing the maximum current density. This was achieved through two specific methods: (1) Reducing the active area from 0.02 mm² to 0.005 mm². This improved the -3dB bandwidth through decreased capacitance (and hence decreased RC time constants). (2) Employing silicon substrates (instead of glass). The superior thermal conductivity of silicon enables device operation at high repetition rates under intense current densities of approaching ~10³ A cm⁻². The frequency response of our laser devices with different active areas and current densities is shown in Extended Data Fig. 10 of the revised manuscript. The maximum modulation bandwidth of the laser devices was improved by 3.6 times, from 10.1 MHz (in our original manuscript) to 36.2 MHz. Further improvements may be possible by increasing the carrier mobilities and concentrations in the perovskite and charge-transport layers.

To address this point, the additional experiments and analyses on the emission process have been provided in Supplementary Fig. 8, Supplementary Fig. 9 and Extended Data Fig. 10, with relevant discussions on page 8 and Supplementary Note 3 (highlighted) of the revised manuscript.

Extended Data Fig. 10 | Frequency response of the electrically-driven dual-cavity perovskite lasers. a, Frequency response characteristics of dual-cavity devices with different active areas. **b,** Frequency response characteristics of a dual-cavity device (active area: 0.005 mm²) at different current densities.

Point 9: There is an inconsistency about the value of the lasing threshold of the dual cavity system, that in the abstract is reported to be 92W/cm² while in the Lasing Reporting Summary is 129 A/cm² (Point 9).

Response: We appreciate the reviewer’s comments. 92 A cm⁻² corresponds to the lowest lasing threshold of our dual-cavity perovskite devices, while the average threshold is 129 A cm⁻² with a standard deviation of 27 A cm⁻². These are stated in point 8 of the lasing reporting summary.

To address this issue, we have now clarified in point 8 of the reporting summary that the minimum and average thresholds are 92 A cm⁻² and 129 A cm⁻², respectively. The information is also given on pages 1, 8 and 9 (highlighted) of the revised paper.

Other comments:

- The manuscript could improve by stating clearly from the beginning that the single crystal microcavity is optically pumped by the PeLED. I suggest to move the sentence “The EL from the microcavity PeLED (microcavity I) was absorbed by the FAPbI₃ single crystal whose emission was then amplified in the second microcavity (microcavity II).” from Methods section “Fabrication of the DBRs” (page 9, lines 251- 253) to somewhere in page 2 in the main text.

Response: We are grateful to the reviewer for the valuable suggestion. To improve clarity, we have moved the sentence from the Methods section to the main text on page 2 (highlighted) of the revised manuscript.

- Given the broad readership of Nature, authors could consider to add to Fig. 1 a sketch describing the working mechanism of the device (similarly to Fig. 4 A).

Page 2, line 65: crystalization -> crystallization

Page 11, line 305: x-ray -> X-ray

Page 11, line 322: integration sphere -> integrating sphere

Response: We appreciate the informative and professional peer review. Following the reviewer's advice, we have now modified Fig. 1b to clarify the working mechanism of the devices. The typos pointed out by the reviewer have been corrected in the revised manuscript.

Referee #2

The authors present a perovskite laser device consisting of vertically integrated dual-cavity structure. One of the cavities is a light-emitting diode, which provides optical power into the other cavity (VCSEL). In contrast to previous studies proposing the similar device architectures, i.e. vertically-integrated LED/VCSEL [ref. 10 and a very recent paper reported from the authors' group (<https://doi.org/10.1126/sciadv.adr8826>)], the current study demonstrates more efficient power delivery owing to the cavity structure enabling the directional optical output. As a consequence, the authors achieve a current-injected perovskite laser module for the first time. The quality of data and presentation is extremely high. Experimental results are presented with appropriate use of statistics and uncertainties are removed.

The novelty of this study, the authors claim, lies in the dual cavity structure. This is certainly true. Furthermore, the authors' group is one of the world's leading groups in perovskite LEDs, and the high-power LEDs presented in this study are likely the result of the authors' maximum use of their technology. However, this study can be viewed as nothing more than a combination of existing technologies, LED microcavity (e.g. <https://doi.org/10.1063/1.116858>) and the integrated LED/VCSEL architecture (ref. 10), and in my opinion, academic progress of this study, as it stands, is insufficient for Nature. Specifically, the authors demonstrate an electrical modulation of their device, but unfortunately, this may expose weaknesses in the authors' device architecture. This is because the modulation rate may be limited by the spontaneous emission lifetime that is the intrinsic operation mechanism for LEDs. This is in contrast to the real LDs ($> \sim$ GHz), which utilize the stimulated emission. If the authors can address this concern and demonstrate that their device is superior in terms of modulation speed (or otherwise), it would provide one important scientific advance.

Response: We are grateful to the reviewer for the encouraging remarks about the quality of data and presentation of our paper, and for acknowledging our efforts in bringing together the technologies we developed in our lab over the past years to create the dual-cavity perovskite laser device. We are equally grateful for your critical comments, which guided us to make important clarifications on the novelty of the work, to improve the modulation performance of the devices to new limits, and to identify future research directions.

Next, we discuss with the reviewer regarding the novel contributions of our work over the former studies. Our paper reports the first demonstration of electrically-driven lasing from

perovskite semiconductors, showing a low lasing threshold of down to $\sim 92 \text{ A cm}^{-2}$, an order of magnitude lower than that of state-of-the-art electrically-driven organic lasers (*Nature* 621, 746-752, 2023). Besides, the electrically-driven perovskite lasers show potential advantages in stability and reproducibility over the organic lasers. The key to the low threshold is the unique dual-cavity architecture, which compactly integrates a low-threshold single-crystal perovskite microcavity sub-unit with a high-power microcavity perovskite LED sub-unit in a vertically stacked multi-layer structure. We believe that our demonstration of electrically-driven lasing from the dual-cavity perovskite devices represents a milestone in the field of perovskite lasers.

We appreciate the reviewer's important remark on the modulation rates, suggesting a direction of improvements for the perovskite laser devices. The overall rise and fall times of the devices are on the order of tens of nanoseconds (Supplementary Fig. 9b). This indicates that the overall frequency response is largely determined by the capacitance-resistance (RC) constant of the microcavity PeLED sub-unit, whose spontaneous emission lifetime is found to be less than 2 ns under intense current injection (Supplementary Fig. 8). Inspired by the reviewer's comments, we have carried out further experiments to improve the modulation bandwidth. This was achieved through two specific methods: (1) Reducing the active area from 0.02 mm^2 to 0.005 mm^2 . This improved the -3dB bandwidth through decreased capacitance (and hence decreased RC time constants). (2) Employing silicon substrates (instead of glass). The superior thermal conductivity of silicon enables device operation at high repetition rates under intense current densities of approaching $\sim 10^3 \text{ A cm}^{-2}$. The frequency response of our laser devices with different active areas and current densities is shown in Extended Data Fig. 10 of the revised manuscript. The maximum modulation bandwidth of the laser devices was improved by 3.6 times, from 10.1 MHz (in our original manuscript) to 36.2 MHz. Further improvements may be possible by increasing the carrier mobilities and concentrations in the perovskite and charge-transport layers. Development of directly-injected perovskite laser diodes is expected to raise the modulation rates toward the GHz range.

Despite our experimental efforts that represent an initial demonstration, we do not anticipate the modulation bandwidths of electrically-driven perovskite lasers to rival the state-of-the-art laser diodes based on III-V semiconductors in the very near future. We acknowledge that the modulation rates are currently a limitation of the devices, and are an important area of future work. We believe that the electrically-driven perovskite lasers, as a new light source, have the following potential advantages: (1) the ability to integrate with silicon-based photonic platforms without employing complex epitaxial processes, and (2) the potentially scalable fabrication of microscale lasers at low costs for wearable and flexible applications.

To address this point, we have improved the modulation bandwidth of the laser devices by 3.6 times, from 10.1 MHz to 36.2 MHz (Fig. 4 and Extended Data Fig. 10). Extended Data Fig. 10 and Supplementary Fig. 8 have been included, with related discussions provided on page 8 (highlighted) of the revised manuscript.

Extended Data Fig. 10 | Frequency response of the electrically-driven dual-cavity perovskite lasers. a, Frequency response characteristics of dual-cavity devices with different active areas. **b,** Frequency response characteristics of a dual-cavity device (active area: 0.005 mm^2) at different current densities.

Supplementary Fig. 8 | Transient PL decay curves at different current densities for a microcavity PeLED with an active area of 0.02 mm^2 . The microcavity PeLED was excited by a femtosecond laser (excitation wavelength: 400 nm ; pulse duration: $\sim 270 \text{ fs}$, excitation fluence: $\sim 1 \mu\text{J cm}^{-2}$), while driven at constant current densities of 5 A cm^{-2} , 8 A cm^{-2} and 10 A cm^{-2} . The spontaneous emission lifetime of the microcavity PeLED is about 2 ns .

Other minor points raised are as follows.

(Definition of ‘cavity’)

- Especially for Microcavity 1, experimental data demonstrating that the optical output is the cavity mode are missing. Does the output from Microcavity 1 before integration with Microcavity 2 show angular dependence of the cavity mode?

Response: We thank the reviewer for raising this important point. The output from microcavity I (the microcavity PeLED sub-unit) shows a clear angular dependence of the cavity mode (Extended Data Fig. 5a&c), without the integration with microcavity II. The EL intensity of the microcavity PeLED sub-unit in the normal direction (0°) is ~ 1.5 times that of a standard PeLED. For the microcavity PeLEDs, the EL spectra blueshift and rapidly decline in intensity at larger angles. Such emission behaviours are very different from those of standard PeLEDs, in which the EL intensity distribution follows the Lambertian profile (Fig. 2f) with angle-insensitive spectral peaks (Extended Data Fig. 5b&d).

To address this point, we have now presented the angle-dependent emission spectra of the PeLED sub-unit in Extended Data Fig. 5, with relevant discussions on page 4 (highlighted) of the revised manuscript.

Extended Data Fig. 5 | Angle-dependent EL of PeLEDs. a-b, Angle-resolved EL spectra of (a) a microcavity PeLED and (b) a standard PeLED. **c-d,** EL spectra measured at 0° , 10° , 20° , 30° , and 40° from the surface normal of (c) a microcavity PeLED and (d) a standard PeLED.

- I understand that the top reflector of Microcavity 1 is the semi-transparent Au. If so, the authors need to show that the Au film exhibits sufficient reflectivity as a VCSEL.

Response: We appreciate the comments. The semi-transparent Au film exhibits a reflectance of 45% at 680 nm (Supplementary Fig. 2). When paired with DBR1 (reflectance: $\sim 99\%$ at 680 nm), it is sufficient for enabling the microcavity effect, as detailed in our response to the previous point of the reviewer's comments. We would like to clarify that the semi-transparent

Au electrode (together with DBR1) enables more concentrated optical power delivery from microcavity I to microcavity II, but it does not provide optical feedback as in a VCSEL structure. Instead, the two DBRs (DBR2 and DBR3) in microcavity II show high reflectance (>98%), providing the optical feedback required for lasing in a VCSEL-like structure.

To address this point, the reflectance spectrum of the semi-transparent Au electrode has been provided in Supplementary Fig. 2, with further details discussed on page 2 (highlighted) of the revised manuscript.

Supplementary Fig. 2 | Reflectance spectrum of the semi-transparent Au electrode in microcavity I.

- What is the physical cavity length in Microcavity 1? In general, perovskite LEDs are composed of emissive and carrier injection layers with a very small thickness (< ~100 nm). If the cavity length is too small, a cutoff condition is imposed on the standing waves in the microcavity. The authors have to exclude this concern.

Response: Thanks for the valuable suggestion. To determine the cavity length of microcavity I, we refer to the cross-sectional STEM image in Fig. 1(b). The functional layers within microcavity I (between DBR and Au) are ITO (~110 nm), ZnO (~20 nm), Cs_{0.5}FA_{0.5}PbI₂Br perovskite (~25 nm), TFB (~30 nm), and MoO₃ (~7 nm). The refractive indices of these layers are ~1.68, ~2.0, ~2.4, ~1.7 and ~2.3, respectively (*Adv. Opt. Mater.* 6, 1800667, 2018). The optical cavity length inside the microcavity (L_{in}) could be obtained using

$$L_{in} = \sum n_i d_i$$

where n_i , d_i are the refractive index and thickness of each layer inside the microcavity. Using this equation, L_{in} is calculated to be 352 nm, which is close to $\lambda_0/2$ ($\lambda_0 = 680$ nm). Such a cavity length is sufficiently long for the emission at ~680 nm.

Further, we consider the possibility of forming an extended cavity, taking into account the penetration of optical fields into the DBRs. The penetration distance into the DBRs (D_{DBR}) can be estimate using (*ACS Photonics* 6, 1804-1811, 2019)

$$D_{DBR} = \frac{\lambda_0}{2} \frac{n_1 n_2}{n_c (n_2 - n_1)}$$

where n_1 and n_2 are the refractive indices of the two alternating materials in the DBR. n_c (1.83) is the effective refractive index inside the cavity. Using this equation, D_{DBR} is calculated to be 697 nm. n_{DBR} is the effective refractive index of the DBR and it can be obtained through

$$n_{DBR} = \frac{n_1 d_1 + n_2 d_2}{d_1 + d_2}$$

where d_1 and d_2 are the thicknesses of the two alternating materials in the DBR. Using this equation, n_{DBR} is calculated to be 1.88.

The total extended cavity length (L_{ext}) equals the cavity length inside the cavity (L_{in}) plus the length of optical penetration into the DBRs.

$$L_{ext} = L_{in} + n_{DBR} D_{DBR}$$

Using this equation, L_{ext} is estimated to be 1662 nm, which is close to $\frac{5}{2}\lambda_0$. Therefore, constructive optical interference at $\lambda_0 = 680$ nm can be established with such a cavity configuration, regardless of whether the effective cavity length is considered to be within the cavity (L_{in}) or extended into the DBR (L_{ext}). The cavity mode at around 680 nm can be supported by the microcavity.

To clarify this point, we have provided the details of the cavity length calculations for microcavity I in Supplementary Note 1, with related discussions on page 2 (highlighted) of the revised manuscript.

- Just the confirmation, Are DBR 2 and 3 based on the same design?

Response: We thank the reviewer for raising this important point. DBR2 and DBR3 are not exactly based on the same design. They both provide high reflectivity (>98%) for the emission (~790 nm) from FAPbI₃ single crystals, allowing effective optical feedback and light amplification in microcavity II. But only DBR2 is required to provide high transmittance (>90%) of the EL (~680 nm) from microcavity I, for the effective excitation of the FAPbI₃ single crystals in microcavity II. In contrast, DBR3 provides low transmittance (<2%) for the EL (~680 nm) from microcavity I. This prevents the emission of microcavity I from escaping from the top surface of the dual-cavity device. As a result, from the output lasing spectrum of the dual-cavity device, we did not observe any residual emission at ~680 nm. In summary, for microcavity II, DBR2 features high reflectance (>98%) at 750-1000 nm, while DBR3 shows high reflectance (>98%) at 620-870 nm.

To address this point, we have provided the reflectance spectrum of DBR3 in Supplementary Fig. 3, with the related information on page 4 (highlighted) of the revised manuscript.

Supplementary Fig. 3 | Reflectance spectrum of DBR3.

(Hyperbole)

- In line 104 (page 4), the term “the highest power output” is misleading as the authors’ LED has a footprint as small as 0.02 mm² the result is in the form of radiance. It is desirable to accurately claim that power can be delivered with high density.

Response: We are grateful for the suggestion. To make a more accurate description, we have revised the term to “the highest output power density” on page 4 (highlighted) of the revised manuscript.

- In Fig. 3e (page 6), It is not fair to compare performance with current injected devices (ref. 9) as the physics that gives the operating principle is very different from the authors’ device.

Response: Thanks for the valuable comments. The purpose of having Fig. 3e was to put our dual-cavity perovskite laser in the context of related technologies, as this might help the general readers and researchers from other fields to better understand our study. In light of the reviewer’s comments, to make a more informative comparison, we have now included more detailed description in the figure to clarify that the organic laser reported in ref. 9 was based on direct injection, while the organic laser in ref. 10 and our perovskite laser both employ integrated pumping structures.

- The authors have very recently published a similar device structure. That study needs to be cited (<https://doi.org/10.1126/sciadv.adr8826>).

Response: We appreciate the suggestion. We have now cited our recent work as ref. 25 (formerly cited as ref. 26 in the original manuscript).

Referee #3

I co-reviewed this manuscript with one of the reviewers who provided the listed reports.

Response: We express our gratitude to the reviewer and their colleague for the constructive comments.

Referee #4

Zou et al. report an electrically-driven halide perovskite laser diode via the same indirect pumping scheme recently used to realize organic laser diodes. The devices implement a DBR microcavity with a single crystal FAPbI gain medium on top of a metal-DBR CsFAPbBrI perovskite LED and show a clear intensity threshold, linewidth narrowing, spontaneous polarization, and increased directionality at threshold.

With appropriate clarification in regard to the questions below, I believe the authors have successfully demonstrated lasing from their devices. Though not particularly novel from a fundamental standpoint given the similarity to the approach taken in Ref. 10, this is certainly an important step forward for the perovskite LED/lasing field and is likely to stimulate more work on indirectly-pumped perovskite laser diode devices in the future. From this standpoint, I think this manuscript merits consideration by Nature, though the following questions/suggestions should be addressed before any decision can be made:

Response: We are grateful to the reviewer for the encouraging and insightful comments, which have provided valuable instructions for us to prepare the revised paper.

1. The authors should clearly show the below threshold spectra in Fig. 3a so that readers can clearly see the spectral collapse for themselves. Simply magnifying the spectra vertically and indicating the magnification factor, e.g. 'x50' or similar designation, would address this. Same concern for the spectra in Ext. Data Fig. 6.

Response: We appreciate the comments. Following the suggestion, we have now included the vertically magnified below-threshold spectra in Fig. 3a and Extended Data Fig. 7 (Extended Data Fig. 6 in original manuscript), with the related information provided in the captions of Fig. 3a and Extended Data Fig. 7c of the revised manuscript.

Fig. 3 | a, Emission spectra of the perovskite laser driven by different current densities. The intensities of the emission spectra below the threshold, at 56 and 85 A cm⁻², are magnified by 10 times and 4 times, respectively.

Extended Data Fig. 7 | c, Emission spectra under different pump intensities. The intensities of the emission spectra below the threshold are magnified by the factors specified near the curves.

2. Given that this is a VCSEL, what breaks the polarization degeneracy above threshold and leads to quasi-linear polarization in Fig. 3c? On a related note, why is the laser emission only weakly polarized? Are there multiple transverse modes lasing, or are there different areas of the DBR/single crystal cavity that are lasing simultaneously? Do the authors observe the same polarization behavior under optical pumping? One way to address this would be to tightly focus the optical pump spot (e.g. $\sim 10 \mu\text{m}$ diameter) and scan it laterally to see whether the polarization changes locally. As it stands, the spot size of the optical pumping beam in Ext. Fig. 6 should be stated.

Response: We thank the reviewer for the valuable comments. The reviewer is correct that in a planar VCSEL-type device structure with perfect axial symmetry, the lasing output should not exhibit preferential orientations of linear polarization (*ACS Photonics* 4, 2486-2494, 2017).

However, it is difficult in practice for such a device to achieve ideal symmetry and isotropy. Structural imperfections, defects and anisotropy are usually present, leading to the breaking of polarization degeneracy. Such polarization characteristics were observed in VCSELs based on a wide variety of gain media, including III-V semiconductors (*Nature* 443, 409-414, 2006; *Opt. Express* 24, 15546-15553, 2016), perovskites (*Adv. Mater.* 29, 1604781, 2017; *Nanophotonics*, 12, 2133-2143, 2023; *Opt. Lett.* 50, 702-705, 2025; *Adv. Opt. Mater.* 9, 2001982, 2021; *Appl. Phys. Lett.* 120, 121104, 2022; *ACS Photonics* 4, 2486–2494, 2017), organics (*Adv. Funct. Mater.* 30, 2003294, 2020), and transition-metal dichalcogenides (TMDs) (*Nano Lett.* 21, 3331-3339, 2021).

In our perovskite devices, lasing output with quasi-linear polarization was observed, possibly owing to minor thickness variation, material anisotropy, formation of strain during crystal growth and DBR deposition, and asymmetric crystal structures (*ACS Photonics* 4, 2486-2494, 2017; *Adv. Mater.* 29, 1604781, 2017; *Nature* 443, 409-414, 2006; *Appl. Phys. Lett.* 77, 1590-1592, 2000; *Opt. Lett.* 50, 702-705, 2025). The conformal coating process could transfer the surface roughness from the underlying perovskite crystals to the DBRs, resulting in structural non-uniformity across multiple interfaces (*Adv. Mater.* 29, 1604781, 2017; *ACS Photonics* 4, 2486–2494, 2017). Concurrently, the residual stress generated during the crystal growth and DBR deposition may introduce anisotropic strain distribution that break the inherent optical isotropy of the system (*Applied Physics Letters* 77, 1590-1592, 2000). The combined effect of structural asymmetry and material anisotropy is expected to create a preferential polarization direction for light emission (Fig. 3c).

We agree with the reviewer that multiple transverse mode lasing and the spatial variations of lasing outputs may lead to weakened polarization. For our devices, as single-mode lasing is observed (Fig. 3a), the effects of multiple transverse modes become negligible. As discussed earlier, the structural non-uniformity across the device leads to local variations in the polarization properties. The moderate polarization observed likely arises from an average effect. To examine such a possibility, following the reviewer's advice we conducted the emission polarization experiments for different regions in microcavity II under optical pumping, using a focused pump spot (~18 μm diameter). Different orientations of linear polarization and DOP values (56% ~ 71%) were observed at different lasing spots (Supplementary Fig. 5). In the electrically-driven dual-cavity devices, the excitation area on microcavity II was about 0.02 mm^2 . In these devices, moderately polarised lasing was observed with a degree of polarization (DOP) of ~54% (Fig. 3c). The orientations of linear polarization of the electrically driven lasing from our dual-cavity perovskite devices also exhibit device-to-device variation (Supplementary Fig. 6).

Supplementary Fig. 5 | Polarization plots for different lasing spots in microcavity II under optical pumping. The lasing spectra were measured with a rotational polarizer. The peak lasing intensity as a function of the polarizer angle is plotted. DOP refers to degree of polarization. Microcavity II was pumped by 1- μ s optical pulses from an electrically modulated 405-nm continuous-wave (CW) laser at a repetition rate of 10 Hz. The excitation intensity was 13 W cm⁻² (above threshold). The diameter of the focused pump spot was \sim 18 μ m.

Supplementary Fig. 6 | Polarization plots of electrically-driven dual-cavity perovskite lasers, showing device-to-device variations in the orientations of polarization. The devices were driven at a current density of \sim 280 A cm⁻² (above threshold).

To address this point, the additional figures are presented in Supplementary Figs. 5 and 6, with relevant discussions on page 7 (highlighted) of the revised manuscript. The spot size of the optical pumping beam has been specified in Extended Data Fig. 7.

3. The authors attribute the lower threshold of their dual cavity device relative to the single cavity to the enhanced directionality of the LED in the former. However, my guess is that the lower threshold has much less to do with the small increase in directionality from Fig. 2f and

much more to do with the higher coupling distance due to the 300 μm thick glass substrate in the latter that effectively results in a 350 μm thick coupling layer (this should be more clearly indicated in Ext. Fig. 7a), which provides more distance for the light to disperse (e.g. the coupling distance is nearly twice the device dimension). The authors should provide an accurate account of how much the increased coupling distance vs. change in LED directionality actually contributes to the lower threshold of the dual cavity device.

Response: We are grateful to the reviewer for the insightful comments. We agree with the reviewer that the inter-cavity coupling distance also plays an important role in reducing the lasing thresholds. To gain insights into the effects of inter-cavity distance and emission directionality, we have simulated the optical coupling efficiencies at various inter-cavity distances for both Lambertian and directional emission (Supplementary Fig. 7a-b). For longer inter-cavity distances, the coupling efficiency is more strongly affected by the emission directionality. At a relatively long coupling distance of 350 μm , simulation results suggest that the inter-cavity coupling efficiency is improved from 17.1% (with a standard bottom-emission PeLED) to 30.2% (with a microcavity top-emission PeLED) (1.8-fold enhancement). This coupling efficiency is further enhanced to 82.7% by reducing the coupling distance from 350 μm to 50 μm (2.7-fold enhancement).

In contrast, for shorter distances, the coupling efficiency is less affected by directionality. For instance, when the distance reduces to 50 μm , directional emission only contributes to an improvement of 1.06 times relative to that of standard Lambertian emission. On the other hand, a 4.8-fold enhancement in coupling efficiencies (from 17.1% to 78.1%) can be achieved by reducing the distance from 350 μm to 50 μm , indicating that reducing distance becomes more effective for devices with more compact vertical integration. The aforementioned simulation was done for devices with an active area of 0.02 mm^2 . If the device area continues to reduce, the benefit of directional emission is expected to become more important (Supplementary Fig. 7c-d). The analysis above suggests the possibilities of further device optimization.

To address this point, a quantitative analysis of the effects of coupling distance and emission directionality is presented in Supplementary Fig. 7, with relevant discussions provided on page 8 of the revised manuscript (highlighted). The coupling distance (350 μm) for the devices featuring standard bottom-emission PeLEDs is now clarified in Extended Data Fig. 9a (Extended Data Fig. 7a in the original manuscript).

Supplementary Fig. 7 | Effects of emission directionality and inter-cavity distance on coupling efficiency between microcavity I and microcavity II. **a**, Simulated coupling efficiency versus coupling distance for Lambertian and directional emission. **b**, The enhancement of the coupling efficiency due to emission directionality for different distances. Device area: 0.02 mm². **c**, Simulated coupling efficiency versus active area for Lambertian and directional emission, at a fixed coupling distance of 50 μm. **d**, The enhancement of the coupling efficiency due to emission directionality for different active areas, at a fixed coupling distance of 50 μm.

4. What is the distance between the laser and the detector in the far-field beam profile measurement in Ext. Fig. 5? Again, please adjust the scaling so that readers can tell for themselves what the FWHM of the curve in panel (b) is.

Response: We thank the reviewer for the valuable suggestions. The distance between the laser and the detector in the far-field beam profile was about 5 cm, as stated in the caption of Extended Data Fig. 6 (Extended Data Fig. 5 in the original manuscript). Following the reviewer’s advice, we have now adjusted the scaling in panel (b) so that the readers can estimate the FWHM of the beam profile. The FWHMs of the beam profiles below and above the threshold are 2.52 mm and 1.18 mm, respectively.

Extended Data Fig. 6 | Far-field beam profiles and line-intensity profiles of electrically-driven perovskite lasers, measured at 5 cm distance from the device. **a**, Far-field beam profile and **b**, the corresponding line-intensity profile (blue curve, magnified by 40 times) measured at 85 A cm^{-2} (below threshold) (the red curve represents the gaussian fit). **c**, Far-field beam profile and **d**, the corresponding line-intensity profile (blue curve) measured at 280 A cm^{-2} (above threshold) (the red curve represents the gaussian fit). The scale bars in **a** and **c**: 1 mm. The measurements were performed at room temperature in air (temperature: $\sim 22 \text{ }^\circ\text{C}$, relative humidity: $\sim 50\%$). The FWHMs of the beam profiles below and above the threshold are 2.52 mm and 1.18 mm, respectively.

5. Improving the English grammar throughout would help with readability.

Response: We appreciate the comments. The revised manuscript has been carefully proofread for improved clarity.

Response to Review Comments:

Referee #1

The manuscript 2025-01-02269A by Zou and coworkers has been thoroughly revised. The authors have seriously considered the criticisms in the original referee reports and have addressed them with scientifically sound arguments, data and analyses.

It is my opinion that the revised manuscript represents a significant improvement and overcomes the threshold for publication.

Response: We are grateful to the reviewer for the encouraging remarks, and for recommending the publication of our paper in *Nature*.

Referee #2

The authors present a perovskite laser device consisting of vertically integrated dual-cavity structure. They responded appropriately to the recent reviewers' comments and revised their paper almost sufficiently. Specifically, the authors claimed, in response to my earlier comment, that the novelty of their study was that current-injected laser operation was achieved at low threshold. This is certainly true and would be a milestone toward a true current-injected laser device, removing my earlier concerns about the novelty.

It is also commendable that additional experiments on the modulation operation demonstrated that the RC time constant of the LED device is the main obstacle limiting the modulation speed. However, even if the RC issue is completely removed in the future, GHz modulation cannot be achieved as long as the PL lifetime is on the order of 2ns. Because of such physical facts, I have concerns about the authors' conclusion that their dual-cavity perovskite laser has potential for communication applications (the final part of page 8). In fact, tens of MHz would be sufficient for spatial visible light communications, in which case LEDs are used instead of LDs. Could the authors link their modulation experiments to other applications? Or, could the authors show the potential for modulation operation of their high-power LEDs in the GHz range? I believe this point is an important one that increases the liability value of this paper.

Response: We thank the reviewer for the insightful comments, and for recognizing the novelty of and importance of our work.

The field of electrically-driven perovskite lasers is still in its infancy. We appreciate the reviewer's remarks on the limited modulation speed achievable by our devices. We agree that achieving GHz operation is challenging, considering the relatively long spontaneous emission

lifetime (~2 ns) of the microcavity PeLED sub-unit. The non-ideal modulation speed is a limitation of our devices, and this constitutes a crucial area of future research.

While the electrically-driven perovskite lasers we developed in this work currently operate at tens of MHz, they still hold potential for data transmission applications in which GHz operation is not essential. Examples of these include radio-over-fiber (RoF) links (*Journal of Lightwave Technology* 28, 2456, 2010), optical wireless data transmission (e.g., Li-Fi) (*Applied Optics* 58, 4553, 2019), and Internet of Things (IoT) device interconnects (*IEEE Internet of Things Journal* 9, 24466, 2022). In these applications, lasers are expected to offer several key advantages over LEDs, such as superior directionality, longer transmission distance and stronger anti-interference capability.

To address this point, we have updated the related discussions on page 6 (highlighted) of the revised manuscript (page 8 of the original manuscript). The abstract, introduction and conclusion sections have been updated, with the original term “communications” replaced with “data transmission” (highlighted).

Referee #3

I co-reviewed this manuscript with one of the reviewers who provided the listed reports.

Response: We appreciate the reviewer and their colleague for their valuable feedback on our work.

Referee #4

The authors have addressed my concerns.

Response: We express our gratitude to the reviewer for providing a supportive and insightful peer review of our paper.